# Multilayer brain networks can identify the epileptogenic zone and seizure dynamics

**Hossein Shahabi[1]\*, Dileep R Nair[2], Richard M Leahy[1]**

[1]Signal and Image Processing Institute, University of Southern California, Los Angeles, United States; [2]Epilepsy Center, Cleveland Clinic Neurological Institute, Cleveland, United States

**Abstract** Seizure generation, propagation, and termination occur through spatiotemporal brain networks. In this paper, we demonstrate the significance of large-scale brain interactions in high-frequency (80–200Hz) for the identification of the epileptogenic zone (EZ) and seizure evolution. To incorporate the continuity of neural dynamics, here we have modeled brain connectivity constructed from stereoelectroencephalography (SEEG) data during seizures using multilayer networks. After introducing a new measure of brain connectivity for temporal networks, named multilayer eigen-vector centrality (mlEVC), we applied a consensus hierarchical clustering on the developed model to identify the EZ as a cluster of nodes with distinctive brain connectivity in the ictal period. Our algorithm could successfully predict electrodes inside the resected volume as EZ for 88% of participants, who all were seizure-free for at least 12 months after surgery. Our findings illustrated significant and unique desynchronization between EZ and the rest of the brain in the early to mid-seizure. We showed that aging and the duration of epilepsy intensify this desynchronization, which can be the outcome of abnormal neuroplasticity. Additionally, we illustrated that seizures evolve with various network topologies, confirming the existence of different epileptogenic networks in each patient. Our findings suggest not only the importance of early intervention in epilepsy but possible factors that correlate with disease severity. Moreover, by analyzing the propagation patterns of different seizures, we demonstrate the necessity of collecting sufficient data for identifying epileptogenic networks.

**\*For correspondence:**
shahabi.h27@gmail.com

**Competing interest:** The authors declare that no competing interests exist.

## Editor's evaluation

This valuable work proposes new network-based algorithms for brain seizure characterisation that could improve the effectiveness of existing clinical treatment paradigms. The approach is supported by solid evidence. If validated and compared against existing biomarkers, it could shed light on mechanisms of disease progression. This work will be of interest to clinicians and researchers in epilepsy alike.

## Introduction

Investigating brain connectivity in epilepsy has attracted considerable attention (***Kramer and Cash, 2012***) since multiple different networks are involved in this neurological disorder (***González Otárula et al., 2019***; ***Kramer et al., 2010***; ***van Diessen et al., 2013b***). Large-scale epileptogenic networks consist of cortical and subcortical areas that are involved in seizure generation and propagation (***Bartolomei et al., 2017***). These networks can either represent an increase or decrease in brain synchrony (***Khambhati et al., 2017***; ***Kramer et al., 2010***; ***Wendling et al., 2003***; ***Yaffe et al., 2015***) or elucidate regions traversed by ictal propagation which is referred to as traveling waves in studies with microscopic recordings (***Martinet et al., 2017***; ***Muller et al., 2018***; ***Proix et al., 2018***; ***Schevon***

et al., 2012; Smith et al., 2016; Weiss et al., 2013). This underlying association between brain areas in epilepsy can describe the seizure spread and termination processes (Kramer and Cash, 2012), explain seizure semiology (Chauvel and McGonigal, 2014), or assist in identifying the seizure onset zone (SOZ) (Burns et al., 2014). Traditionally, seizures have been considered to be characterized by a state of hypersynchrony. Yet recent work (Kramer and Cash, 2012) describes an early stage of desynchronization (Kramer et al., 2010; Schindler et al., 2010; Wendling et al., 2003) followed by synchronization amid seizure termination (Schindler et al., 2007; Schindler et al., 2008). Constructing connectivity maps with various recording techniques and different computational measures over a wide range of frequencies has given rise to different controversial perspectives on how seizures should be characterized (Jiruska et al., 2013). Few studies have examined the functional connectivity between SOZ and other areas of the brain during ictal periods. Electrocorticographic (ECoG) data suggested that for some patients the SOZ is isolated from the rest of the network early in the seizures allowing for SOZ detection in this way (Burns et al., 2014). It has also been suggested that ictal periods can be delineated by a steady series of states (Burns et al., 2014), although whether this is true in all patients remains controversial. Others have shown a decreased synchrony between SOZ and normal brain regions (Warren et al., 2010). It remains unclear how the degree of desynchronization is correlated with physiological parameters such as age and duration of epilepsy (van Diessen et al., 2013a).

Several studies have used multiunit recordings to explore ictal propagation networks (Martinet et al., 2017; Schevon et al., 2019; Schevon et al., 2012; Smith et al., 2016; Weiss et al., 2013). At the microscopic spatial scale, the hypersynchronous ictal core with high neural firing can be distinguished from the penumbra with relatively small and sparse firings (Schevon et al., 2012). In the early part of the seizure, the slow-moving ictal wavefront involves the core and surrounding areas. After recruitment, the low-frequency traveling waves rapidly propagate to other cortical regions (Smith et al., 2016) in a two-dimensional spatial scale (Martinet et al., 2017). However, the mechanism by which ictal discharges spread in the brain volume is still undetermined. The intense firing of neurons in the ictal core is characterized by high-frequency oscillations (HFOs) (Jefferys et al., 2012) in local field potentials (LFPs) (Weiss et al., 2013). The term HFOs has been attributed to brain activity, with multiple possible physiological and pathological neural mechanisms, between 80–500 Hz (Jacobs et al., 2012) or 30–600 Hz (Engel and da Silva, 2012). This includes high-gamma neural activities (80–200 Hz) (Ray and Maunsell, 2011) and broad-band high frequency (Arnulfo et al., 2020).

These oscillations have been analyzed for their value in SOZ localization (Höller et al., 2015; Liu et al., 2018) during ictal (Weiss et al., 2013) and interictal (Fedele et al., 2017; Gliske et al., 2018) periods. Nevertheless, manual HFO (ripple) detection is time-consuming (Gliske et al., 2018) and automatic approaches produce a large number of false positives (Bénar et al., 2010). Moreover, some interictal analyses of slow-wave sleep questioned the utility (Roehri et al., 2018) and accuracy (Jacobs et al., 2018) of HFOs to serve as a biomarker for epileptogenic tissue identification. These observations can be further supported by a resting-state SEEG study which has delineated the long-range high-frequency synchronization of physiological HFOs among the non-epileptogenic regions (Arnulfo et al., 2020). While HFOs have been employed in analyzing functional (Schindler et al., 2010) and propagation (González Otárula et al., 2019) networks, the spatiotemporal dynamics of ictal high-frequency synchronization (HFS) at macroscopic scales remain largely unknown.

In this paper, we analyzed SEEG recordings of cortical and subcortical regions. In SEEG, the EZ not only takes into account the earliest ictal EEG change, it emphasizes an anatomo-electro-clinical analysis (Kahane et al., 2006). This concept incorporates both the anatomic region that initiates the epileptic discharge as well as the 'primary organization' (Talairach and Bancaud, 1966) that leads to the manifestation of the clinical seizure itself (Wyllie et al., 2015). The gold standard method of confirming EZ localization is based on whether seizure freedom has been achieved by resection or ablation. The actual ground truth for the EZ location is unknown since in many cases the resection volumes may extend well beyond the EZ. In recent work, the EZ has been considered as part of a network (Jehi, 2018). These epileptogenic networks in focal epilepsy have been invoked in explaining the underlying pathogenesis of epilepsy, seizure initiation, ictal propagation, and disease progression as well as various associated comorbidities (Nair et al., 2004). This perspective is utilized in our work to analyze seizures in the context of a distributed network of interacting regions that include the EZ. Because of the importance of HFOs in epilepsy and SOZ localization, we construct synchronization networks in the 80–200 Hz range, to be in line with similar studies (Höller et al., 2015; Schindler

*et al., 2010*). Fast rhythmic bursting neurons, which have the highest tendency to initiate seizures, are largely responsible for generating ultra-fast oscillations or ripples (80–200 Hz) (*Timofeev and Steriade, 2004*). We hypothesize that during seizures, the EZ has an abnormal and unique pattern of connectivity with other brain areas.

Graph analysis provides a mathematical framework for the quantification of brain connectivity (*Bullmore and Sporns, 2009*). Brain networks may be represented as a graph, $G = (V, E)$, in which nodes ($V$) characterize anatomical regions or electrodes, and edges ($E$) reflect structural or functional connections among them. Traditionally, brain connectivity in each time sample (layer) of dynamic networks has been evaluated independently, via single graph analysis. However, multilayer analysis allows us to model the entire data with a single super-graph, in which individual graphs for each time-sample are linked. Assuming there are $T$ single graphs (time-samples) with $N$ nodes in each, the super-graph has $NT$ nodes. The multilayer structure has considerable methodological advantages over single-layer analysis (*Betzel and Bassett, 2017*). First, the interlayer coupling between neighboring time points in this model allows us to incorporate the continuity in neural dynamics. Second, by tuning the coupling parameter, processes with different timescales can be distinguished. Third, the extracted measures on these networks are less susceptible to noise in the data or spurious connectivity. Additionally, there exist several neurological rationales for employing a multilayer approach in seizure analysis. First, the concept of dynamic network reconfiguration has been studied in brain networks (*Bassett et al., 2011*; *Braun et al., 2015*). Previous research has shown state transitions during seizures, either through brain connectivity analysis (*Burns et al., 2014*) or microelectrode recordings (*Smith et al., 2016*). Multilayer networks have the capability to delineate these transitions and identify network reconfiguration (*Mucha et al., 2010*). Second, electrophysiological signals are highly non-stationary during ictal periods. As a result, traditional analysis of time-varying networks based on isolated graphs would be affected by instantaneous fluctuations rather than the underlying spatiotemporal networks. Third, seizure propagation is one of the key elements of ictal activity. Recent work has indicated the importance of multilayer modeling of complex systems when encountering spreading processes (*De Domenico et al., 2016*).

Consequently, we modeled spatiotemporal high-frequency connectivity using multilayer networks. We explore the question of whether the EZ can be identified by unsupervised clustering of nodes (representing SEEG contacts) in the feature space of these multilayer networks. Our connectivity-based EZ identification results show reasonable consistency with a previous approach (described as the fingerprint of the EZ) (*Grinenko et al., 2018*; *Li et al., 2020*) which uses three ictal features for EZ localization, namely: low-voltage fast activity (LFD), preictal spiking, and suppression of lower frequencies.

Understanding the connectivity dynamics of EZ and surrounding areas with the rest of the brain can shed light on underlying processes including seizure propagation and termination. We evaluate these interactions in different frequency bands. To use consistent terminology, here the term 'high-frequency synchrony' is mostly utilized to describe brain networks at higher frequencies (80–200 Hz). On the other hand, the word 'connectivity' has been assigned to both propagation networks in low frequency (3–50 Hz) and synchronization networks in high frequency (80–200 Hz). Our findings illustrate early high-frequency desynchronization and late increase in brain connectivity during seizures. Although, seizures usually initiate with a loss of synchronization in a small area, their termination process demands widespread brain connectivity across multiple scales. During seizure cessation, the brain experiences a critical transition with a hysteresis behavior (*Kramer et al., 2012*), indicating the future state depends on the current one. Accordingly, we hypothesize that post-termination high-frequency synchrony is correlated with pre-termination connectivity in low frequency.

## Results

### Multilayer modeling of ictal networks discerns the EZ

SEEG data were recorded using implanted intracranial electrodes in 16 patients who underwent resective surgery and were seizure-free for at least 12 months post-resection (*Table 1*). We studied the dynamics of brain connectivity during seizures via multilayer networks (*Mucha et al., 2010*) which captures continuity in neural interactions during the ictal period. A schematic of a multilayer network with inter and intralayer edges is depicted in *Figure 1*. SEEG contacts were defined as graph vertices

**Table 1.** Clinical characteristics of patients.

| ID | Age (years) | ED (years) | MRI lesion | Resection/ablation details | Surgical pathology | Follow-up (months) | Anatomical location of the EZ | Number of nodes in the network | Number of nodes inside the resection area | Duration of seizures (seconds) |
|---|---|---|---|---|---|---|---|---|---|---|
| 1 | 43 | 37 | FCD, insular/frontal operculum | Anterior insular/frontal operculum | FCD type 2B | 13 | Insular/frontal operculum | 88 | 11 | (41, 39, 39) |
| 3 | 33 | 17 | Hippocampal sclerosis | Anterior temporal lobe | Hippocampal sclerosis | 48 | Temporal | 79 | 22 | (147, 150, 141) |
| 4 | 17 | 8 | Negative | Laser ablation, superior frontal gyrus | No pathology | 19 | Frontal | 71 | 5 | (25, 24, 25) |
| 5 | 16 | 1 | Benign neoplasm, posterior para-hippocampal gyrus | Posterior para-hippocampus gyrus and neoplasm | Low grade glial/glioneuronal neoplasm | 39 | Basal posterior temporal | 48 | 8 | (55, 115, 140) |
| 6 | 46 | 41 | FCD, mesial frontal | Prefrontal lobe | Non-specific | 38 | Frontal | 88 | 32 | (100, N/D, N/D) |
| 7 | 5 | 1 | Negative | Superior frontal gyrus, superior frontal sulcus, frontal pole | FCD type 2B | 21 | Superior frontal gyrus/superior frontal sulcus | 73 | 33 | (14, 15, 15) |
| 8 | 63 | 14 | Negative | Orbitofrontal | FCD type 1 | 44 | Orbitofrontal/ pars orbitalis | 105 | 19 | (61, 267, 62) |
| 9 | 33 | 19 | Gliotic postoperative changes | Anterior temporal lobe | FCD type 1B | 40 | Temporal | 99 | 46 | (85, 64, 81) |
| 10 | 21 | 11 | Negative | Occipital lobe | Gray matter heterotopia, FCD type 1B | 12 | Cuneus | 123 | 57 | (106, 98) |
| 11 | 32 | 27 | FCD, precentral gyrus | Precentral gyrus | Non-conclusive | 77 | Precentral gyrus | 82 | 13 | (26, 78, 12) |
| 12 | 22 | 3 | FCD, superior frontal sulcus | Superior and middle frontal gyri, anterior cingulate | FCD type 2B | 78 | Frontal | 58 | 31 | (18, 25, 31) |
| 13 | 19 | 18 | Negative | Middle frontal gyrus | FCD type 1 | 48 | Inferior frontal sulcus/middle frontal gyrus | 41 | 26 | (36, 36, 35) |
| 14 | 30 | 18 | Negative | Frontal operculum | FCD type 2B | 47 | Frontal operculum/ subcentral region | 70 | 10 | (49, 21, 78) |
| 15 | 20 | 11 | Negative | Frontal lobe | FCD type 1 | 82 | Superior frontal gyrus/superior frontal sulcus | 99 | 32 | (65, 86, 86) |
| 16 | 65 | 25 | Negative | Anterior temporal lobe | FCD type 1 C | 39 | Temporal | 139 | 23 | (56, 65, 142) |
| 17 | 65 | 9 | Negative | Anterior temporal lobe | FCD type 1 C | 36 | Temporal | 90 | 35 | (55, 63, 59) |

Follow-up information is current as of July 2017.

ED: epilepsy duration, FCD: focal cortical dysplasia, N/D: Not defined.

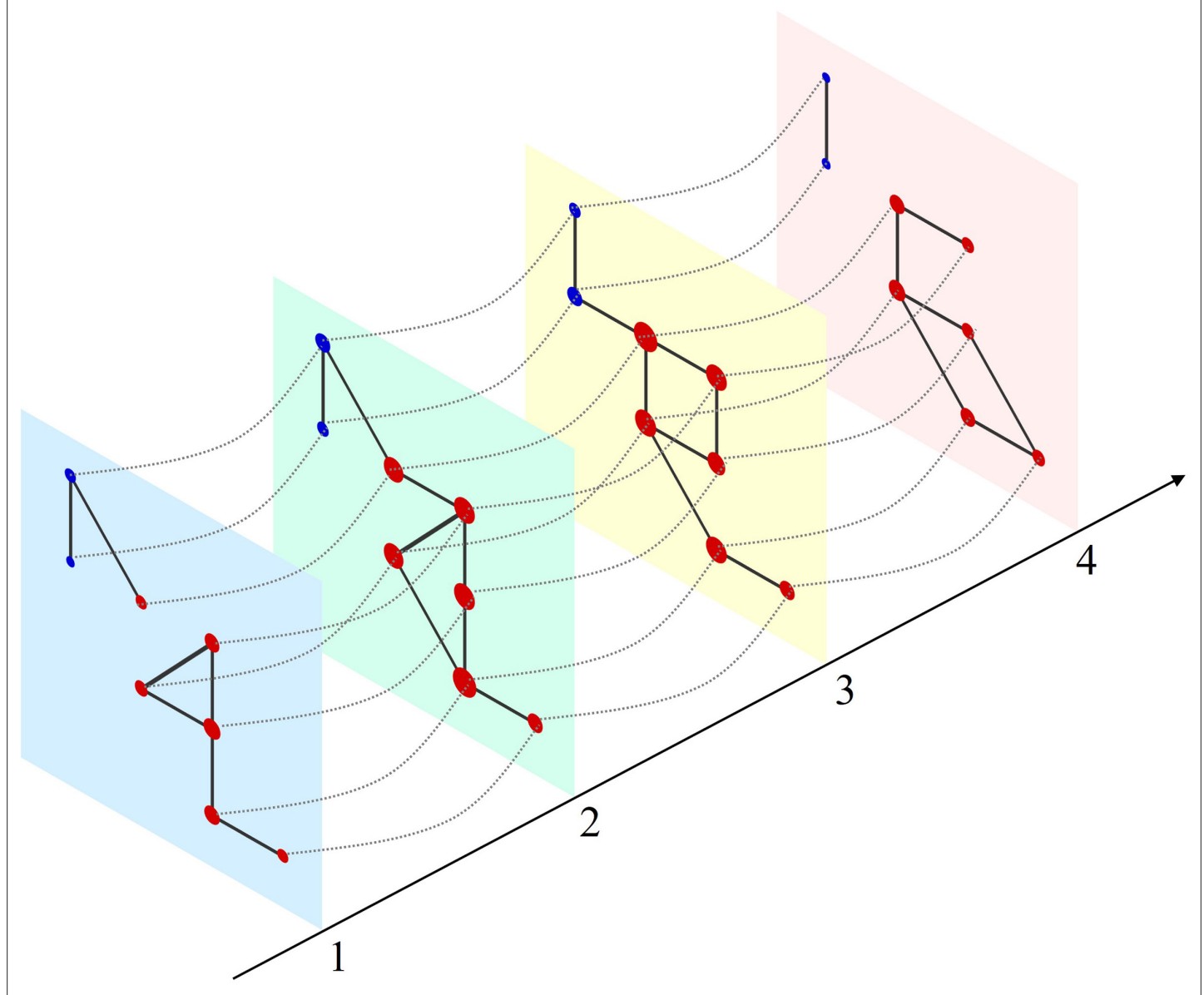

**Figure 1.** Schematic of a multilayer network and measure of multilayer eigenvector centrality (mlEVC). Four consecutive layers (time-points) of a simulated network with intralayer (solid lines) and interlayer (dashed lines) edges. Nodes in each layer represent the set of stereoelectroencephalography (SEEG) contacts and are colored and categorized into two clusters (red and blue). Interlayer edges (couplings) were included between the same contacts at adjacent time points. The diameter of each node represents the relative value of mlEVC, in which larger nodes are the more connected nodes and smaller nodes are more isolated.

while the lagged-coherence (*Pascual-Marqui, 2007*) was used to define edge strength in each layer as a measure of the macroscopic HFS (see Methods). We investigated two broad bands of HFOs: 80–140 Hz and 140–200 Hz, similar to related studies (*Arnulfo et al., 2020*; *Weiss et al., 2013*). Several studies have applied phase-based connectivity metrics on broad-band high-frequency oscillations. For instance, (*Zweiphenning et al., 2016*) computed the phase lag index (PLI) in two high-frequency bands: ripple (80–250 Hz) and fast ripple (250–500 Hz). Another study used PLI to compute brain connectivity in broad band 80–250 Hz (*Nissen et al., 2016*). Last, *Burns et al., 2014* used coherence to investigate ictal networks in gamma (25–90 Hz) band.

As discussed in the introduction, we aimed to model ictal brain connectivity by multilayer networks. Additionally, we were interested in quantifying network dynamics using centrality metrics (*Rubinov and Sporns, 2010*). Among those measures, eigenvector centrality (EVC), a rank-based

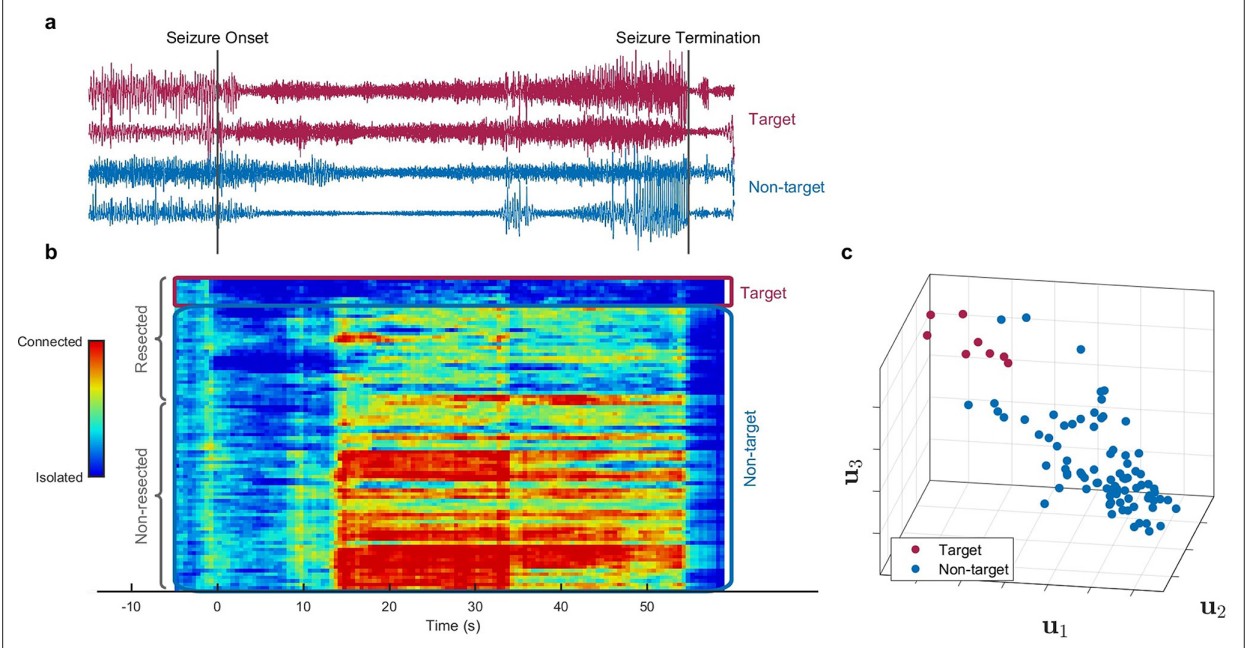

**Figure 2.** Identification of epileptogenic zone (EZ) based on multilayer eigenvector centrality (mlEVC) (Patient 17). (**a**) Stereoelectroencephalography (SEEG) signals during the ictal period. The red signals are a sample of contacts inside the EZ as identified by our method, and the blue signals depict non-EZ contacts outside the resection zone. (**b**) The mlEVC during the ictal period. Each row represents a channel (contact). Contacts are categorized and organized into two groups: resected and non-resected as indicated on the right. They are also categorized into two groups based on the proposed clustering algorithm: target and non-target. The blue elements of mlEVC describe the isolated nodes of the super-graph (brain network) while red values describe highly connected contacts. (**c**) The first three left singular vectors of the mlEVC. The mlEVC spectra of different seizures were first quantized and concatenated before performing singular value decomposition. The red nodes are those identified as predicted EZ based on unsupervised clustering.

The online version of this article includes the following source data and figure supplement(s) for figure 2:

**Source data 1.** Comparing SEEG channels identified as true positives between the two methods, mlEVC in this paper vs. those identified using the Fingerprint in *Grinenko et al., 2018*.

**Source data 2.** This table summarizes the prediction performance for the proposed method and Fingerprint algorithm on 16 patients.

**Source data 3.** This table compares the prediction performance of the proposed method with a single-layer network and a multilayer network with a fixed coupling parameter for all subjects.

**Figure supplement 1.** Epileptogenic zone (EZ) localization for 16 patients.

**Figure supplement 2.** Comparison of multilayer eigenvector centrality (mlEVC) for original (left) and phase-randomized (right) seizure data for a representative subject.

**Figure supplement 3.** Comparing time-frequency spectrums between nodes in epileptogenic zone (EZ) and non-resected regions.

**Figure supplement 4.** Magnitude of eigenvalues in multilayer graphs with different coupling parameters.

metric that assesses the importance of each node in the network (*Bonacich, 1972*), has been employed in studying seizures (*Burns et al., 2014*). Mathematically, the leading eigenvector of the adjacency (connectivity) matrix has been assigned as the EVC of the graph when there is one clique (component) in the matrix (*Bonacich, 1972*). However, in our multilayer model of time-varying brain connectivity, a single vector cannot explain the complex structure of the spatiotemporal networks. Therefore, we introduced a new measure called mlEVC that incorporates the top T eigenvectors of the adjacency matrix of the super-graph (see Methods). This allows us to evaluate patterns of nodal centrality and identify regions with similar connectivity characteristics to the rest of the graph. Further, because mlEVC is a function of the interlayer coupling parameter (*c*), we can explore neural processes at different timescales by varying *c*. The mlEVC represents the prominence of a node in multiplex networks evolving over time (*Figure 1*). Nodes with high connectivity over time and space in the multilayer network display larger values in mlEVC than isolated vertices. *Figure 2b* displays this measure for one seizure of patient 17.

## Algorithm for predicting EZ using mlEVC

We hypothesize that the EZ can be identified as the set of nodes in the graph that exhibit a characteristic and distinct pattern of connectivity to other areas during the seizure. To explore this question, we first quantized the mlEVC of each seizure into three levels based on percentile thresholding ($d$). The top $d/2$-portion of elements was assigned a value of '1,' the bottom $d/2$-portion a value of '–1,' and the remainder a value of '0.' For each subject, quantized measures of mlEVC for two high-frequency bands and all seizures were concatenated to a single matrix with dimension $N$ by $T_{tot}$ (twice the total number of sample points) (see Methods). We applied the singular value decomposition (SVD) to the concatenated matrix and used the left singular vectors ($u_i \in R^N$) to identify nodes (SEEG channels) with similar features. As an illustration, **Figure 2c** depicts $u_1$ , $u_2$, and $u_3$ for patient 17.

Next, we applied an unsupervised clustering algorithm using the $u_i$ vectors to detect a target cluster that represents the EZ. Following our initial hypothesis, the target cluster should portray a dense and distinctive set of nodes in the feature space with a significant distance from nodes in the non-target group (**Figure 2c**). We describe the clustering algorithm in detail in the method section. Briefly, we designed a data-driven framework to cluster nodes into two groups using an agglomerative hierarchical clustering technique (function *linkage* in MATLAB). This process was performed for different combinations of $u_i$ as features of the clustering algorithm and a range of values for $c$ and $d$, resulting in 440 clustering runs. We weighted each run using a performance function (**Halkidi et al., 2001**) that examines the tightness of the target cluster and its separation from other nodes. Finally, using the weighted sum of performance metrics for all runs, SEEG contacts were divided into two groups; target and non-target.

## Assessing the accuracy of the prediction algorithm

By comparing the clustering results with information about which contacts were included in the resected volume (**Figure 2b**), we defined three categories: 'EZ,' 'resected non-EZ,' and 'non-resected.' The EZ included the nodes in the target cluster, resected non-EZ consists of nodes in the non-target cluster removed during surgery, and non-resected comprises the rest of the nodes, which were neither resected nor clustered in the target group (**Figure 2—figure supplement 1**). Ideally, the predicted EZ or target cluster should only contain nodes in the resected area for these participants since all patients were seizure-free after surgery. However, the clustering algorithm and proposed technique are not flawless so there are a small number of electrodes outside the resection region selected as EZ, i.e., false positives.

Our approach identified electrodes inside the resected volume as EZ for 88% of participants, i.e., all but two (patients 8 and 11). For patient 11, 2 out of 3 available seizures were short (25 s and 12 s) and the sampling rate was relatively low (500 Hz). Lack of sufficient samples might be the source of miss identification for this patient and a limitation of our approach. The false-positive rate (FPR) was calculated by dividing the number of predicted electrodes as EZ outside the resection zone over the total number of electrodes in the non-resected region. Only four participants had electrodes falsely identified as EZ and the FPR was 1.79% across all 16 patients. The details of predicted electrodes for each patient are given in **Figure 2-source data 1**. We compared our results with Fingerprint (**Grinenko et al., 2018**) analysis that employs time-frequency features of pre- and post-seizure onset, to predict the EZ. Although these two algorithms are fundamentally different in their assumptions, contacts/nodes identified in our approach as EZ have 41% overlap (same contact labels) with those found using the Fingerprint. Interestingly, the mlEVC algorithm could identify electrodes inside the resection zone for two patients in which Fingerprint was not able to predict the EZ (participants 6 and 16). In contrast, the latter could localize EZ for two patients (8 and 11) on which our method failed. For several patients, the two approaches identified different but adjacent areas inside the resection region. Nonetheless, 67% of patients had common electrodes labeled as EZ in the resected volume by both algorithms. Since both analyses were retrospective and the actual ground truth is unknown, these techniques can perhaps be used to complement each other. (see **Figure 2-source data 1** and **Figure 2—source data 2** for a detailed comparison). Finally, **Figure 2—source data 3** demonstrates that the proposed method utilizing weighted consensus clustering outperforms single-layer networks and multilayer networks with fixed coupling.

It was striking that for most of the patients, the EZ was distinguishable based on the singular vectors of mlEVC (**Figure 2—figure supplement 1**). Consistent with other studies, we found out that

possibly only a portion of the resected area is responsible for epileptogenicity. *Figure 2b* captures the discrepancies between the EZ, resected non-EZ, and non-resected areas. Furthermore, it distinctly displays various brain states (phases) during the ictal period. Nodes experience isolated and fully connected phases with respect to the rest of the network. These results confirm the importance of the entire ictal period for EZ identification. For the rest of the paper, we utilize the categorization of nodes introduced above with the following small modifications: removing false positives from the EZ groups (defined as those contacts lying outside the resected volume) and discarding patients 8 and 11 in whom we were unable to observe any unique pattern of connectivity in electrodes inside the resected area.

## Evaluating the validity of the proposed method

To explore the validity of our approach, we constructed a null model by also computing the mlEVC from phase-randomized SEEG signals (*Prichard and Theiler, 1994*). In *Figure 2—figure supplement 2*, we show typical results of the mlEVC measures calculated from the original time-series and phase-randomized data for a single subject. These results show significant differences between original and randomized data in which the characteristic patterns of brain connectivity both in resected and non-resected areas are lost when signals are phase-randomized. We performed this analysis for different patients and seizures with similar findings.

To investigate whether apparent high-frequency networks were produced as a result of ictal spikes rather than true oscillations, we examined the time-frequency representation of seizures. *Figure 2—figure supplement 3* displays time-frequency plots for two sample seizures. The upper plot shows the spectrum averaged among nodes predicted as EZ and the lower spectrum shows the case for randomly selected nodes outside the resection region, with the number of nodes in both groups equal. We observed pre-ictal spikes in the time-frequency plot for nodes inside the EZ, a previously reported feature of the EZ (*Grinenko et al., 2018*). However, the figure shows that non-spiking high-frequency oscillations were the dominant activity after seizure onset. This observation supports our hypothesis that HFOs, rather than ictal spiking, are the main contributor to observed synchronization during seizures.

Furthermore, we examined the effect of multilayer modeling and adjusting the coupling parameter on EZ prediction and super-graph structure. In *Figure 2—figure supplement 1*, we display the left singular vectors of mlEVC. For illustration purposes, we showed projections with respect to the singular vectors for the parameter set ($c$, $d$) for which the prediction of EZ (binary labeling) was closest to the assumed ground truth (resection information). We observed that the optimal coupling value is different between patients, which supports our primary rationale for choosing a multilayer framework with an adaptive interlayer coupling parameter. In our initial analysis, we ran the algorithm for the single-layer case and EZ identification results were poor (see *Figure 2—source data 3*).

*Figure 2—figure supplement 4* presents the changes in super-graph eigenvalues by adjusting the coupling parameter. For $c \leq 1$, there is a falloff when the number of eigenvalues ($n_e$) meets the number of layers ($T$), indicating a super-adjacency matrix with effective rank $T$. This is an expected result since the coupling is relatively small. When $c$ increases, the eigenvalues become larger, and their corresponding eigenvectors would comprise several neighboring layers. Although the rate of decline in the magnitude of eigenvalues accelerates as $c$ increases, falloff can be detected when $n_e \leq 2$ $T$. By increasing the coupling value, the super-adjacency matrix transforms from a block diagonal matrix with an effective rank $T$, to a matrix with major non-diagonal blocks and an effective rank far greater than $T$. In the computation of mlEVC, we considered the top $T$ eigenvectors for all coupling parameters to avoid erroneous assumptions about the rank of super-graphs.

## Seizures evolve with divergent network topologies

In our multilayer modeling of epileptogenic networks, the left singular vectors of mlEVC were used to cluster nodes into two categories, predicted EZ and non-EZ. Here, the associated right singular vectors were employed to display seizure evolution. In clinical epileptology, stereotypy in seizures is defined as similarities in both seizure semiology and ictal EEG recordings over repeated seizures (*Schevon et al., 2012*). Multiunit recordings have shown that stereotypical firing patterns occur when micro-electrodes are implanted in recruited areas (*Schevon et al., 2012*). To evaluate stereotypy in this study, we clustered ictal dynamics into different states using the four features from a pool of

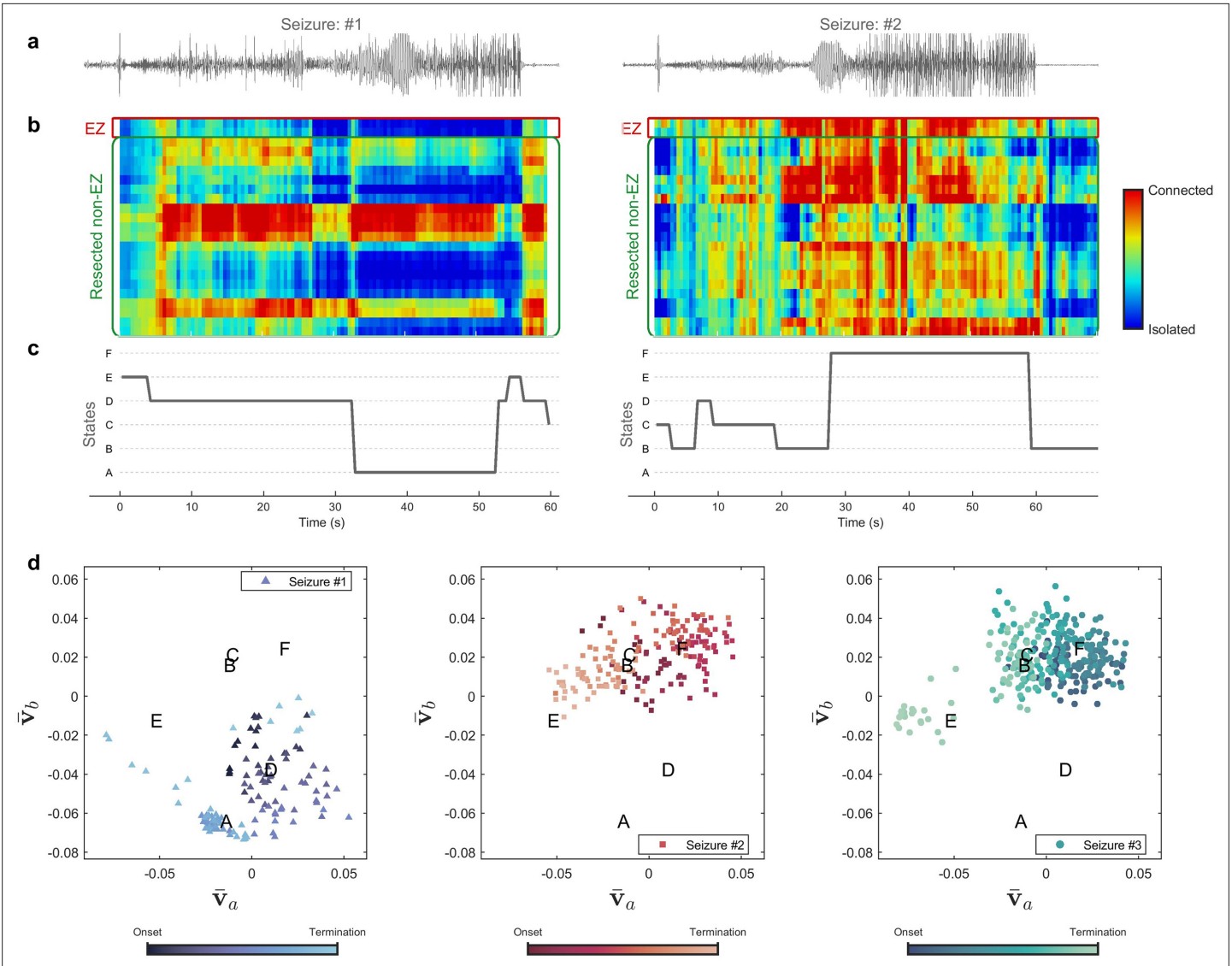

**Figure 3.** Evolution of seizures through high-frequency states (Patient 16). (**a**) Time series of a channel inside the predicted epileptogenic zone (EZ) for two different seizures (onset to termination). (**b**) The low-rank estimation of quantized multilayer eigenvector centrality (mlEVC) plots (80–140 Hz). For a clearer representation, only channels in predicted EZ and resected non-EZ are depicted. (**c**) State transitions during seizures. The time vector is adjusted based on the seizure onset (t=0 s) (**d**) Evolution of seizures in the feature space. Capital letters show the center of each brain state. The space and states are created by four features ($\bar{v}_a$, $\bar{v}_b$, $\bar{v}_c$, and $\bar{v}_d$) extracted from the right singular vectors of mlEVC in the two high-frequency bands (see Methods – Here, only two features are illustrated). Comparing three ictal periods, we see that the brain can exhibit a different seizure evolution – here between seizure one and seizures two and three.

The online version of this article includes the following figure supplement(s) for figure 3:

**Figure supplement 1.** Seizure evolution and state transition in three patients.

six-dimensional feature vectors, including the top three right singular vectors of mlEVC split into the two high-frequency bands (see Methods).

*Figure 3* compares different seizures for patient 16. Quantized mlEVC are depicted in *Figure 3b* with corresponding state changes in part c. Results indicate that ictal brain activity evolves through diverse phases during different seizures. In the first seizure, the EZ is mostly isolated with respect to the other nodes and ictal activity advances through three states (**E, D, A**). In contrast, the EZ exhibits strong connectivity in the second seizure while the centrality measure passes through four states (**C, B, D, F**). *Figure 3d* illustrates the ictal dynamics using the four selected features extracted from the

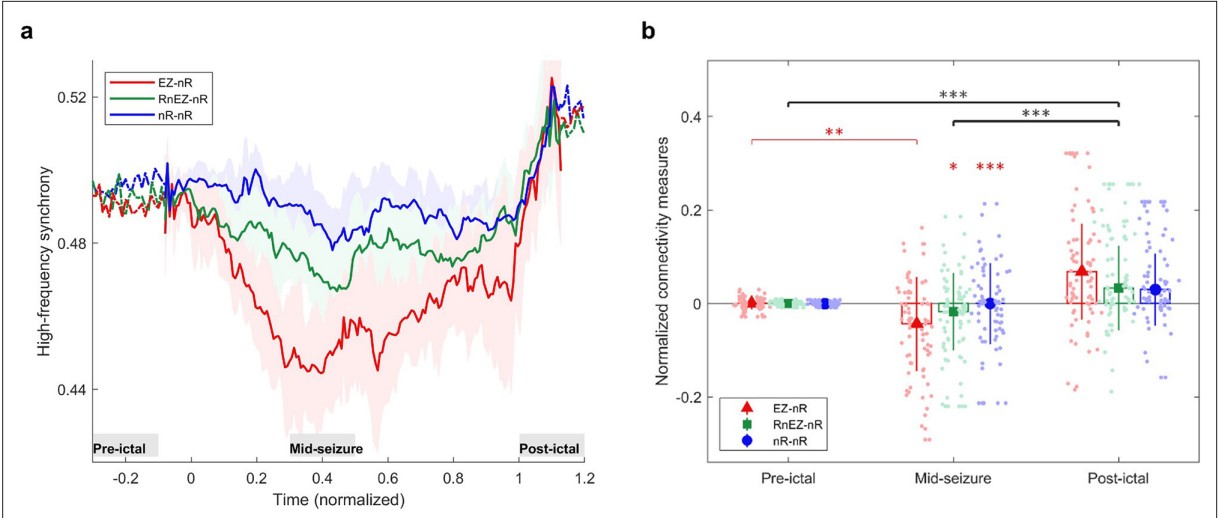

**Figure 4.** High-frequency synchrony during the ictal period. (**a**) Time-varying high-frequency synchrony values for three defined measures. The ictal period is normalized to a zero to one scale. The solid lines represent the median and shaded plots display the normalized median absolute deviation (MAD), based on $10^4$ bootstrap tests. The gray rectangles display the periods of special interest. Note that EZ-nR connectivity drops substantially towards mid-seizure and increases to match nR-nR and RnEZ-nR at seizure termination and post-ictally. (**b**) Connectivity measures in pre-ictal, mid-seizure, and post-ictal. To give a pairwise visual comparison, we subtracted the average synchrony between all pairs of contacts in the pre-ictal interval from nine connectivity measures. The centers of error bars show the median of all seizures in two frequency bands and lines depict the MAD. Scatter circles exhibit the actual values (n=78 for each group). To handle possible outliers, the paired percentile bootstrap with a one-step M-estimator was employed and p-values were computed for pairwise comparison between and within the three measures at each of the three periods, corrected using Hochberg's algorithm (B=$10^4$ number of bootstraps, J=18 tests, see Methods). Asterisks display corrected p-values; *p<0.05, **p<0.01, ***p<0.001. Only the mid-seizure interval shows significant differences between EZ-nR and other measures. All measures are considerably higher in post-ictal than their corresponding values in the mid-seizure and pre-ictal periods. EZ-nR drops significantly in mid-seizure from pre-ictal.

The online version of this article includes the following source data and figure supplement(s) for figure 4:

**Source data 1.** Corrected p-values for all pairwise comparisons.

**Figure supplement 1.** Connectivity measures in pre-ictal, mid-seizure, and post-ictal.

top three right singular vectors of mlEVC in two frequency bands (see Methods). In this case, seizure one evolves quite differently from seizures two and three whose traces share similar ictal dynamics.

The same analysis was performed for all patients and results for several of these participants are shown in *Figure 3—figure supplement 1*. We observed that stereotypy, here considered as similar state transitions among all recorded seizures, does not necessarily occur, especially at high frequencies. In fact, the brain experiences divergent topologies, which might be the result of dissimilar EEG recordings. While previous studies suggested stereotypy in focal firing (*Schevon et al., 2012*) and brain connectivity (*Burns et al., 2014*), our multilayer analysis of epileptogenic networks does not imply stereotypy in macroscopic HFS. Our findings confirm the necessity of collecting adequate seizures and large-scale recordings for a better understanding of seizure evolution.

## EZ desynchronization occurs in the ictal period

We were interested in exploring the fundamental question of how HFS changes throughout the seizures. To do this, three synchrony measures were computed. The first measure, EZ-nR, quantifies the synchrony between EZ and non-resected (nR) areas. The second metric, RnEZ-nR, computes the connectivity of resected nonEZ and nR. Lastly, we computed interactions between non-resected regions, labeled nR-nR. *Figure 4a* presents the dynamics of these measures during the ictal period. Collectively, 39 seizures and two frequency bands were analyzed. Seizures with different durations were resampled/rescaled to a zero to one interval, where zero indicates the onset and one indicates the termination time. In general, among the three measures, EZ-nR exhibited a substantial decline in synchrony during early and mid-seizure while widespread synchronization occurred during seizure termination (*Figure 4a*). Based on this observation, we compared the HFS in three time periods: pre-ictal, mid-seizure, and post-ictal (*Figure 4a*). Data for all seizures were extracted for statistical

analysis in RStudio (*Rstudio Team, 2018*). To handle possible outliers, the paired percentile bootstrap with a one-step M-estimator (*Wilcox, 2011*) was employed and p-values were computed for pairwise comparison between and within the three measures at each of the three periods, corrected using Hochberg's algorithm (B=$10^4$ number of bootstraps, J=18 tests, see Methods). All tests and their corresponding corrected p-values can be found in *Figure 4—source data 1* .

   *Figure 4b* presents the extracted connectivity values for the above measures in selected periods. To give a pairwise visual comparison, we subtracted the average synchrony between all pairs of contacts in the pre-ictal interval from nine connectivity measures (*Figure 4b*). Unsurprisingly, there was no difference between measures in the pre-ictal period ($p \approx 1$ for all pairwise comparisons). In mid-seizure, EZ-nR was significantly smaller than RnEZ-nR (p=0.0108) and nR-nR (p<$10^{-4}$), indicating that the EZ is maximally desynchronized from the rest of the brain. Early-onset and late-ictal HFOs have been considered biomarkers for seizure onset zone identification (*Weiss et al., 2013*), with the latter found to be a more reliable metric (*Modur et al., 2011*). Our EZ localization technique considers both features. The substantial decrease in EZ connectivity with the entire network in mid-seizure might be the result of these pathological HFOs in the EZ. At a smaller scale, EZ-nR desynchronization could be the result of heterogeneous neuronal spiking activity during seizures (*Truccolo et al., 2011*). Micro-electrode recordings of the ictal core presented a dramatic rise in the Fano factor, a statistical measure of spiking desynchronization (variance of spiking divided by the mean), in the early and mid-phases of seizures (*Schevon et al., 2012*; *Truccolo et al., 2011*). Theoretical modeling of neuronal assemblies has shown that asynchrony is necessary to maintain a high firing rate (*Gutkin et al., 2001*). We further studied the variations of each measure among different periods. Between pre-ictal and mid-seizure, EZ-nR connectivity declined considerably ($p \approx 0.002$), followed by a marginally significant fall for RnEZ-nR (p=0.056). In contrast, all measures were substantially elevated during seizure termination and the post-ictal period in comparison to mid-seizure and pre-ictal intervals (p<$10^{-4}$ for all tests). This observation is well aligned with other studies, suggesting a widespread synchronization during seizure termination (*Kramer and Cash, 2012*), especially in the range 80–200 Hz (*Schindler et al., 2010*).

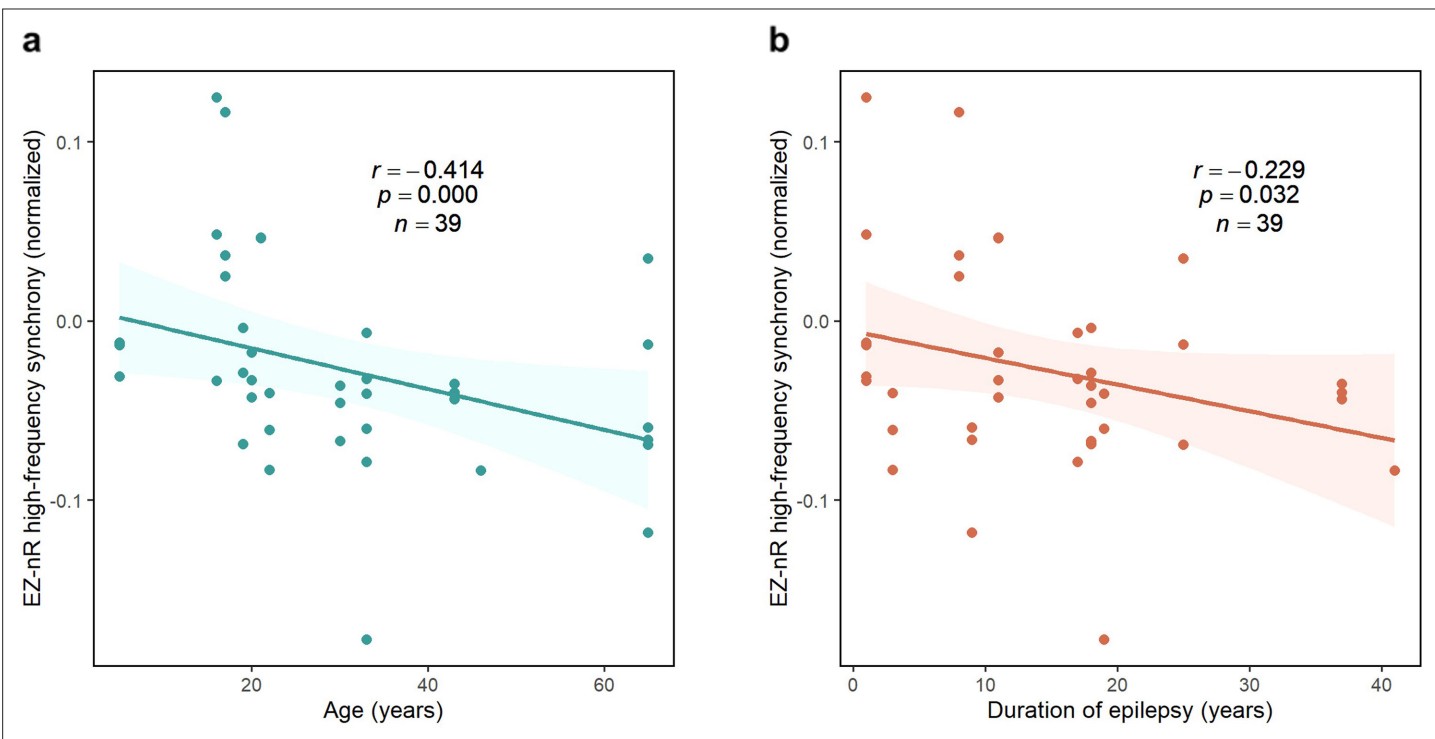

**Figure 5.** Correlation between normalized EZ-nR values and patients' history. (**a**) Correlation between patients' age and normalized EZ-nR synchronization in high frequency. Each data point indicates the average of normalized EZ-nR values in mid-seizure among two high-frequency bands for each seizure (n=39). We utilized a robust regression estimator based on bootstrap sampling and the Theil-Sen algorithm (*Wilcox, 2011*). (**b**) The same plot as (**a**) except where the x-axis indicates the duration of epilepsy among patients.

We do not include the connectivity between the EZ and RnEZ in *Figure 4* for the following reason. The three measures we do compute are all relative to the non-resected region which we know to definitely be outside the EZ. Within the resection, there is some ambiguity as to which contacts are within EZ and which are not. Second, a measure between EZ and RnEZ would be more susceptible to noise as the number of RnEZ electrodes is typically much smaller than the number in the non-resected region so that establishing statistical significance is difficult. Nevertheless, this measure is computed for the interested reader in *Figure 4—figure supplement 1*. It can be perceived by visual inspection that EZ-RnEZ has a pattern similar to EZ-nR, suggesting again that the EZ is functionally disconnected from surrounding areas up to mid-seizure (*Warren et al., 2010*).

## The EZ becomes isolated by aging and the duration of epilepsy

Although the EZ exhibited a general pattern of desynchronization, it was not the case for all patients. In mid-seizure, several participants showed larger EZ-nR values when compared with the average connectivity in the entire network. This observation is expected since patients have dissimilar seizure types, etiology, and electrode implantations. Consequently, we postulated that a patient's demographics might explain differences in EZ connectivity with the rest of the brain. Patients' age and duration of epilepsy were assessed as predictors for the EZ-nR synchrony in the middle of seizures. For each seizure and patient, the average synchrony between all pairs of contacts in mid-seizure was subtracted from the EZ-nR to reduce inter-subject and inter-seizure variabilities. We constructed a three-dimensional vector consisting of EZ-nR connectivity for each seizure (n=39) along with the corresponding age and duration of epilepsy. We utilized a robust regression estimator based on bootstrap sampling and the Theil-Sen algorithm (*Wilcox, 2011*). The correlation between patients' age and normalized EZ synchronization is shown in *Figure 5a*. Results indicate a strong negative association, suggesting the EZ becomes increasingly desynchronized with age ($p<10^{-4}$, $r = -0.414$). Similarly, we observed a reduction in connectivity between the EZ and non-resected areas with a longer duration of epilepsy (*Figure 5b*, p=0.032, $r = -0.229$).

These findings suggest possible variables that can modify the epileptogenic networks (*van Diessen et al., 2013a*). Recently, interictal ECoG recordings of patients with temporal lobe epilepsy (TLE) have shown a negative correlation between TLE duration and overall PLI at low frequencies (*van Dellen et al., 2009*). A resting-state fMRI study also found a negative correlation between epilepsy duration and functional connectivity between two contralateral ROIs in the inferior frontal gyrus (*Liao et al., 2010*). However, our findings delineate the correlation in a specific pathological pathway among patients with medically intractable focal epilepsy with different SEEG electrode implantations. A decrease in functional connectivity with age might also be observed in a control group. However, the fact that we normalize by subtracting the overall synchronization in each patient from EZ-nR values weakens the influence of that factor in our findings.

## Expansive connectivity in low-frequency emerges before seizure termination

Low-frequency brain signals (3–50 Hz) are mainly shaped by rhythmic synaptic currents (*Schevon et al., 2012*), which in many cases traverse to other regions. These traveling waves are involved in different sensory processes and brain states (*Muller et al., 2018*; *Smith et al., 2016*). In epilepsy, ictal discharges exhibit this activity during seizures. Recently, two scenarios have been proposed for seizure spread and termination (*Martinet et al., 2017*). The first theory postulates that ictal discharges emerge from a fixed cortical source in EZ, while the second hypothesis asserts that the moving ictal wavefront generates traveling waves (*Smith et al., 2016*). This process dominates when the ictal wavefront recruits the seizure core and penumbra, i.e., the area around the core in which low-voltage signals spread, roughly during the mid-seizure period. These scenarios have contradictory explanations for how seizure termination occurs. The fixed source theory assumes inactivation of a small region would end the seizure while the active wavefront requires a mechanism that affects an expansive area.

We analyzed connectivity in low-frequency during seizures. Brain networks were constructed using the PLI. The PLI is recognized for its superiority to measures such as phase locking value (PLV). By discarding interactions with a phase difference of zero, the PLI quantifies phase coupling between two signals while excluding volume conduction as a confounding factor. Compared to other functional connectivity metrics, PLI is, therefore, less susceptible to common sources. Since a major part

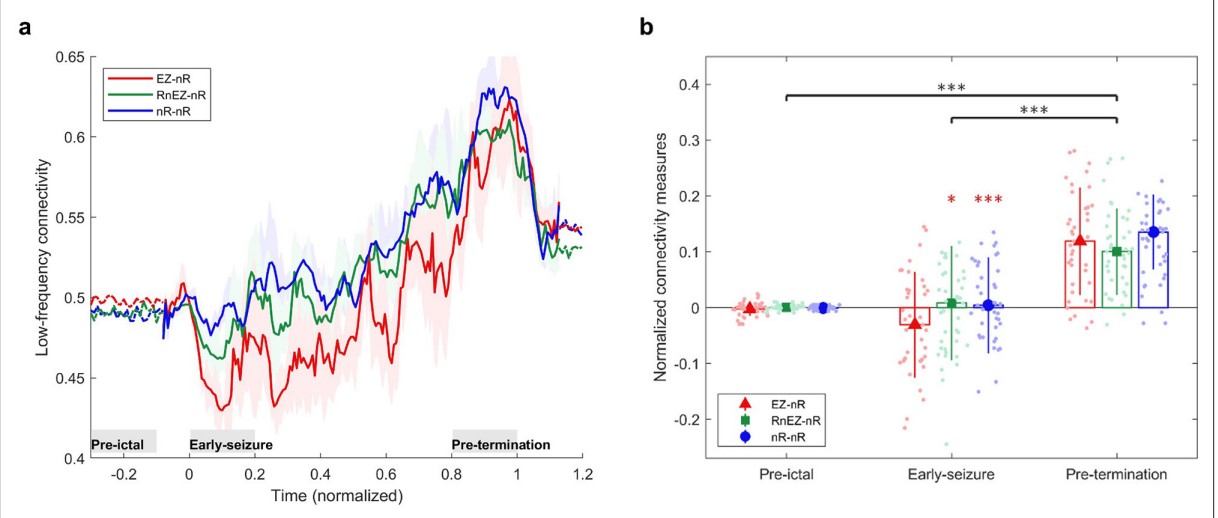

**Figure 6.** Low-frequency brain connectivity during the ictal period. (**a**) Time-varying low-frequency connectivity values for three defined measures. The ictal period is normalized to a zero to one scale. The solid lines represent the median and shaded plots display the normalized median absolute deviation (MAD) based on $10^4$ bootstrap tests. The gray rectangles display the periods of special interest. EZ-nR connectivity drops in early seizure and a widespread brain connectivity occurs in the pre-termination period. (**b**) Connectivity measures in pre-ictal, early-seizure, and pre-termination. To give a pairwise visual comparison, we subtracted the average synchrony between all pairs of contacts in the pre-ictal interval from nine connectivity measures. The centers of error bars show the median of all seizures in two frequency bands and lines depict the MAD. Scatter circles exhibit the actual values (n=39 for each group). To handle possible outliers, the paired percentile bootstrap with a one-step M-estimator was employed and p-values were computed for pairwise comparison between and within the three measures at each of the three periods, corrected using Hochberg's algorithm (B=$10^4$ number of bootstraps, J=18 tests, see Methods). Asterisks display corrected p-values; *p<0.05, **p<0.01, ***p<0.001. Only the early-seizure interval shows significant differences between EZ-nR and the two other measures. All measures are considerably higher in pre-termination than their corresponding values in early-seizure and pre-ictal periods.

The online version of this article includes the following source data for figure 6:

**Source data 1.** Corrected p-values for all pairwise comparisons.

of low-frequency (<50 Hz) brain activity during seizures is derived from ictal discharges, which exhibit distance-dependent delays (*Smith et al., 2016*), true neurological zero-lag connectivity is highly unlikely in this scenario. Because of its robustness to volume conduction, PLI has been widely used in studies of low-frequency (<50 Hz) brain connectivity in epilepsy (*Nissen et al., 2016*; *Schevon et al., 2012*; *van Diessen et al., 2013b*; *Zweiphenning et al., 2016*). Similarly to high-frequency networks, these connectivity matrices were normalized to the pre-ictal period (see Methods). Previously defined measures were computed, and their dynamics are depicted in *Figure 6a*. In comparison to HFS, we observed a stronger resemblance in PLI among the three interaction measures (EZ-nR, RnEZ-nR, and nR-nR) other than in a short early-seizure period. Additionally, ictal discharges displayed an expanding coverage after mid-seizure, which was maximized before seizure termination. As a result, early-seizure and pre-termination along with pre-ictal periods were chosen for statistical analyses as analogs to HFS (n=39, see *Figure 6—source data 1*). We did not find any difference between measures in pre-ictal and pre-termination PLI (p>0.05 for all cases). However, EZ-nR was distinguishable from RnEZ-nR (p=0.0374) and nR-nR (p<$10^{-4}$) in early seizure. This observation can be linked to early-onset suppression in low-frequency, reported as a possible signature of the EZ (*Grinenko et al., 2018*). Compared with early-seizure and pre-ictal periods, all measures were significantly elevated in pre-termination (p<$10^{-4}$ for all tests). Consistent with our findings, a recent study showed that an increased temporal and spatial correlation along with flickering, the condition when a system fluctuates between two attractors, are signatures of a critical transition when seizures self-terminate (*Kramer et al., 2012*).

## Pre-termination connectivity predicts post-ictal synchronization

It has been hypothesized that the brain manifests hysteresis between the two states before and after termination (*Kramer et al., 2012*). In other words, the post-ictal state is dependent on pre-termination. Consequently, we were interested to examine how the brain changes between these

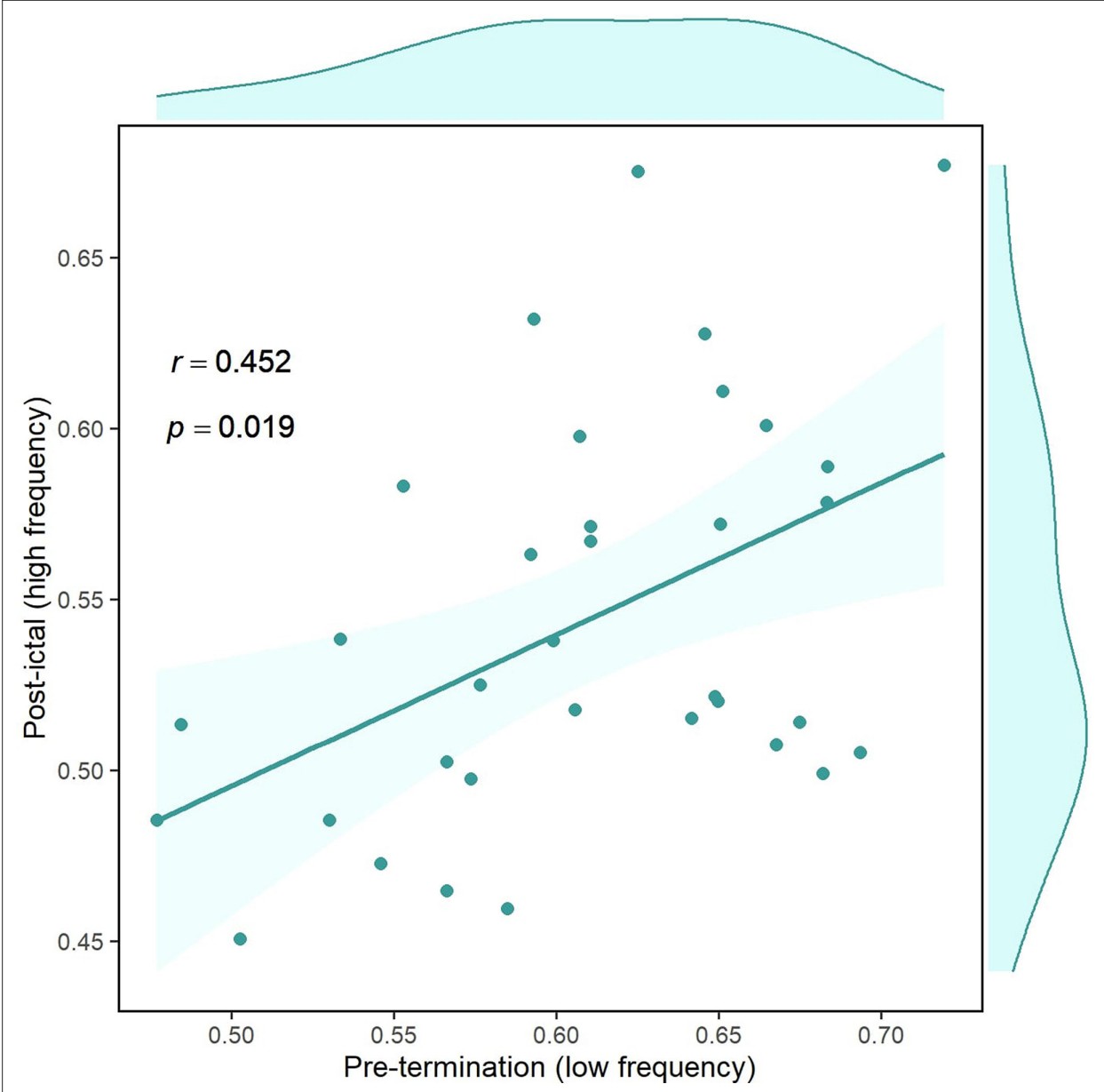

**Figure 7.** Correlation of overall pre-termination low-frequency connectivity and post-ictal high-frequency synchrony during the critical transition. Each point in the graph belongs to one seizure in which the two values for two high-frequency bands are averaged (n=34). Outliers were removed using the projection method and MAD-median rule (*Wilcox, 2011*).

two states. Pre-termination connectivity in low-frequency and post-termination synchrony in high-frequency were among the distinctive features of the results presented above. To test the hypothesis of possible dependency, we computed overall brain connectivity for these two measures. *Figure 7* illustrates the correlation between pre-termination and post-ictal intervals. Each point in the graph belongs to one seizure in which the two values for two high-frequency bands are averaged. There is a strong association between the two states (p<0.02, *r*=0.452, n=34) after removing outliers using the projection method and the MAD-median rule (*Wilcox, 2011*), supporting the existence of hysteresis in the system. In other words, the brain state in the post-ictal period can be predicted using its condition in pre-termination.

## Discussion

Our results investigate large-scale ictal brain connectivity as it relates to EZ localization and seizure generation, propagation, and termination. We showed that the EZ exhibits differential dynamics compared to other brain regions during seizures. This observation is independent of seizure type, etiology, presence or absence of MRI lesion, and occurred in both temporal and extra-temporal lobe epilepsy. High-frequency oscillations have been widely studied as a potential biomarker in epilepsy (*Zijlmans et al., 2012*), including both early- and delayed-onset ictal HFOs (*Weiss et al., 2013*). However, there is still some debate on their spatial specificity, timing, and appropriate detection algorithms. As an alternative marker, we computed high-frequency synchrony (HFS) over the entire seizure period. HFS was first modeled using a multilayer network and the potential EZ was identified using a novel measure of centrality. Our approach to identifying the EZ is complementary to other approaches avoiding explicit assumptions regarding seizure patterns and dynamics.

The multilayer network approach used here for EZ prediction and identification of state transitions allows us to explore seizure dynamics in a graph-theoretic context. Traditionally in brain network studies, researchers explore graph measures like clustering and centrality, in single-layer (non-dynamic) networks (*Ridley et al., 2015*; *van Diessen et al., 2014*). However, the multilayer framework with an adjustable coupling parameter can reveal processes with different timescales and facilitate defining of new measures (*Betzel and Bassett, 2017*). By adjusting the coupling between layers, we can overcome the current challenges of selecting an appropriate time interval between different samples of a temporal network. Here, we assumed a relatively short time interval between layers (500 ms) to have a high temporal resolution. By increasing the coupling parameter between adjacent layers, we can investigate processes with slower timescales.

We used an unsupervised clustering algorithm to find a set of nodes among all channels, i.e., the target cluster, as the predicted EZ. The presented method was based on hierarchical clustering and a cost function that combines the separation of clusters and the compactness of the target group. The optimized coupling parameter varies substantially between patients (*Figure 2—figure supplement 1*), verifying the necessity of analyzing multiple timescales. In some cases, this automatic unsupervised clustering may fail to find all nodes (contacts) with features that would identify them as belonging to the EZ using other methods, such as the fingerprint method that finds a distinctive combination of pre-ictal spiking, low-frequency suppression, and low-voltage fast activity (LFD). This is reflected in the differences in contacts identified as EZ between the two approaches, which is reported in *Figure 2—source data 1*. However, the complementary dynamic-synchrony-based approach described here also finds plausible EZ contacts in cases where the fingerprint does not, as mentioned in the result section. Finally, based on the predicted EZ and information regarding the resection areas, the SEEG electrodes were categorized into three groups; EZ, Resected non-EZ (RnEZ), and non-resected (nR). We employed this categorization to compute the dynamics of regional ictal connectivity in low and high-frequency bands.

There is substantial evidence of multiscale alterations in structural and functional brain networks in epilepsy (*Bernhardt et al., 2016*; *Englot et al., 2016*; *Tavakol et al., 2019*). A meta-analysis of a dozen interictal studies with variant imaging methodologies showed increased clustering and path length (*van Diessen et al., 2014*). Resting-state fMRI research has revealed decreased inter-hemispheric functional connectivity in medial and lateral temporal regions among patients with TLE (*Maccotta et al., 2013*; *Pittau et al., 2012*). Ictal SEEG recordings of focal cortical dysplasia (FCD) type II have shown that nodes within the lesion have higher values of out-degree and betweenness centrality in the gamma range (30–80 Hz)(*Varotto et al., 2012*). Using cortico-cortical evoked potentials (CCEPs), we recently presented the differences in effective connectivity between FCD types I and II (*Shahabi et al., 2021*). These observations have created a new field of research known as connectivity-based biomarkers in epilepsy (*Larivière et al., 2021*), which is mostly performed by automatic approaches and machine learning algorithms. Applications include estimating neurocognitive performance (*Mazrooyisebdani et al., 2020*), lesion detection in FCD type II (*Jin et al., 2018*), lateralization of seizure focus (*Barron et al., 2015*), and ictal onset zone identification (*Van Eyndhoven et al., 2019*). Although these measures have been extracted using non-invasive resting-state imaging techniques, in many cases they are limited to specific types of epilepsy, such as FCD type II or lesional patients. On the other hand, we demonstrated that during seizures nodes belonging to the EZ share a unique pattern of centrality, which was verified in the majority of our patients with different etiologies.

Our results are consistent with iEEG (*Warren et al., 2010*) and ECoG (*Burns et al., 2014*) studies that demonstrated the isolation of SOZ from the rest of the network. More importantly, we investigated the relation between EZ-nR connectivity and patient demographics. We observed a negative correlation between these parameters, in which the EZ and non-resected regions become more desynchronized by aging and the duration of epilepsy (*Figure 5*). This finding can be explained by abnormal neuroplasticity, where the continuous recruitment of epileptogenic networks intensifies their anomaly (*Tavakol et al., 2019*). Related work has shown a negative correlation between TLE duration and overall interictal PLI at low frequencies (0.5–48 Hz) (*van Dellen et al., 2009*). A resting-state fMRI study indicated that the connectivity between two contralateral regions in the mesial temporal lobe is negatively correlated with the duration of epilepsy (*Liao et al., 2010*). Our results along with the studies cited suggest that network abnormalities can portray the disease severity (*Tavakol et al., 2019*). To further verify these findings, additional studies should be performed on a larger dataset while considering other influencing factors, including sex, handedness, and lateralization of the SOZ.

The origin and frequency range of HFOs in ictal and interictal states are still disputable (*González Otárula et al., 2019*; *Jacobs et al., 2012*; *Korzeniewska et al., 2014*; *Tamilia et al., 2018*; *Weiss et al., 2013*). Recent research has emphasized that narrow-band physiological HFOs, not pathological HFOs or broad-band multi-unit activity (MUA), are responsible for long-range high-frequency synchronization in interictal recordings (*Arnulfo et al., 2020*). Also, it has been suggested that widespread HFO synchronization should be a characteristic of healthy brain activity (*Arnulfo et al., 2020*). In our findings, the ictal HFS between the EZ and non-resected areas was the most distinctive pattern among others and showed a significant desynchronization in the early and middle parts of the seizure (*Figure 4*). In this period, the exclusive presence of pathological HFOs inside the EZ resulted in decreased EZ-nR synchrony while the nR-nR connectivity was left intact. Towards seizure termination, this unique activity faded (*Smith et al., 2016*) and elevated the similarity of signals in the EZ and non-resected areas, which is consistent with a previous study in the same frequency range (*Schindler et al., 2010*). Once the seizure terminated and the brain resumed a healthy activity, we observed a widespread HFS irrespective of pathology, a possible outcome of physiological HFOs.

We used the right singular vectors in mlEVC decomposition to explore the network topology during seizures (*Figure 3*). This feature demonstrated the reconfiguration of brain networks and was later used for clustering these dynamics into brain states. The profound alterations of these features revealed the extensive changes in network topology, even when the nR-nR connectivity was relatively stable. Several studies have also displayed the presence of different brain states in the ictal period (*Burns et al., 2014*; *Khambhati et al., 2015*). However, our findings are distinctive in that they present dissimilar brain states among different seizures of the same individual, suggesting variant structures for seizure generation and propagation. It has been shown that TLE seizures can be divided into several categories and in an overwhelming majority of patients they belong to more than one category (*Bartolomei et al., 2010*). Consequently, in each individual, seizures can be generated by various and separate epileptogenic structures (*Bartolomei et al., 2017*; *Bartolomei et al., 2010*). Another study of TLE has identified patients with different seizure semiologies (*Loesch et al., 2015*), a possible outcome of variations in propagation networks. Recently, the Fingerprint features of the EZ were employed for clustering the seizures of each patient. In the presence of these variations in epileptogenic networks, it is essential to record multiple seizures (*Struck et al., 2015*) and prolonged interictal (*Gliske et al., 2018*) data. Also, seizure dissimilarities have been assessed by brain network measures in a wide frequency range (*Schroeder et al., 2020*). Here, we presented a new approach to group seizures based on the features of high-frequency synchronization networks. Additional work can be done to associate these findings with the patient's semiology.

While we mainly focused on the dynamics of high-frequency synchrony, the low-frequency (3–50 Hz) propagation networks were also computed, to have a better understanding of large-scale brain interactions during seizures. Here, the PLI was used to address delays between ictal discharges in different channels. In early-seizure, the EZ-nR PLI was significantly smaller than connectivity among non-resected areas. This finding can be related to early-onset suppression of low frequencies in the EZ, as demonstrated in recent studies (*Aupy et al., 2019*; *Grinenko et al., 2018*). The low-frequency suppression can be described by rapid inhibitory synaptic currents which resist the seizure spread, causing inhibitory restraint (*Schevon et al., 2012*; *Trevelyan, 2009*). By mid-seizure, the ictal wavefront has recruited more areas and ictal discharges are traversing farther regions (*Smith et al., 2016*).

Our results indicated an extensive increase in brain connectivity approaching seizure termination (*Figure 6*). This observation is consistent with multiscale recordings, reporting an increase in spatial and temporal correlation before seizures cease (*Kramer et al., 2012*; *Martinet et al., 2017*), as well as macroscopic cortical networks (*Jiruska et al., 2013*; *Kramer et al., 2010*; *Schindler et al., 2007*). The underlying mechanisms of seizure generation, evolution, and termination remain a matter of debate (*Martinet et al., 2017*). In addition to studies based on microscopic recordings and modeling, macro-electrode studies and a large-scale explanation of brain dynamics are necessary to develop a full understanding of this issue. Our findings suggest that at seizure-onset, the EZ loses synchronization with the rest of the network and the brain enters a desynchronized state. This condition continues through mid-seizure until the ictal wavefront has recruited many areas, including the core and penumbra (*Smith et al., 2016*). Subsequently, brain connectivity increases temporally and spatially, indicating an upcoming critical transition (*Kramer et al., 2012*). It has been shown theoretically and experimentally that brain networks require a higher coupling strength to restore synchronization than to lose synchronization (*Kim et al., 2018*). Our results support this idea since considerably stronger connectivity was needed to terminate seizures in comparison to that in the pre-ictal period.

It is important to remember that the recording techniques, signal processing approaches, and frequencies of interest are crucial in any study of brain connectivity. These parameters can be the reasons for current controversies (*Jiruska et al., 2013*). Recent research has investigated the effect of incomplete electrode sampling on network metrics (*Conrad et al., 2020*), in which EVC showed the highest reliability in comparison to other network metrics when randomly removing network nodes. However, they also showed that confidence intervals in network measures can vary significantly between patients. For this reason, care should be taken when interpreting and comparing different studies.

Employing the bipolar montage and a robust measure of synchrony, helped us to eliminate the effect of volume conduction and muscle artifacts (see Methods). In contrast, classic measures like coherence strikingly increase spurious connectivity. Additionally, our findings do not simply reflect the distance among the electrodes. We normalized the synchronization matrices with respect to the pre-ictal period based on an element-wise approach, previously suggested in *Burns et al., 2014*. This process reduces the chance of distance alone affecting the value of connectivity.

## Limitations and Considerations

We should point out that cross-validation was not performed in this study for several reasons. First, resection labels were not used as part of the prediction algorithm, and the same performance function and ranges of coupling and thresholding values were applied to all subjects. Second, as shown in this manuscript and similar works, parameters such as coupling or active frequency bands are patient-specific. By using cross-validation, we would have to fix these parameters based on a subset of patients, which may not be ideal as the actual range of these parameters could differ from the training dataset. While this study was primarily conducted to propose the multilayer network methodology during seizures, a more rigorous investigation is necessary to evaluate the broader validity of our algorithm. Future research could use data from different clinical centers to assess the generalizability of the proposed technique and identify optimal parameters.

Filtering and subsequent analysis of HFOs should be performed with extra caution due to the possibility of producing false or spurious ripples. Research has shown that high-pass (>80 Hz) filtering of sharp transitions in EEG (such as artifacts and spikes) or harmonics of non-sinusoidal signals can result in such false ripples (*Bénar et al., 2010*). Several approaches have been suggested to investigate this phenomenon. These include visually inspecting the epileptic waveform alongside the filtered signal, using the time-frequency transform of the original signal since real and false ripples have different spectra, and employing automated spectral decomposition techniques such as Matching Pursuit (*Bénar et al., 2010*). In our analysis, we visually evaluated several seizures in both the time and time-frequency domains. However, it would be beneficial to use one of these mentioned algorithms in a systematic manner in future studies.

# Materials and methods

## Patients and recordings

This retrospective study was approved by the institutional review board at the Cleveland Clinic. Single pulse electrical stimulation-induced cortico-cortical evoked potentials are collected as a part of the routine clinical care of patients undergoing SEEG at Cleveland Clinic. The ictal data is also collected during the presurgical SEEG evaluation. The full procedure for participant selection and data recording is described in our previous work (*Grinenko et al., 2018*). Briefly, we selected 16 patients who underwent SEEG implantation in the Epilepsy Center at Cleveland Clinic. SEEG placement (*Gonzalez-Martinez et al., 2014*) used multi-lead depth electrodes (AdTech, Integra, or PMT). Post-implanted 3D computed tomography (CT) images were aligned to T1-weighted MRI for anatomical localization of electrode leads. Patients were monitored for up to two weeks and their seizures were recorded by the Nihon Kohden EEG system with a sampling rate of 500 Hz (before 2012) or 1000 Hz (after 2012). After a thorough evaluation, the identified EZ was resected or ablated. The contacts (electrode leads) inside the resection or ablated region were determined by coregistration of the post-implant CT to a post-resection MRI 1–6 months after surgery. Based on follow-up information, all patients were determined to be seizure-free (*Table 1*).

We preprocessed SEEG signals before further analyses. First, we constructed bipolar channels by subtracting unipolar signals recorded from each pair of adjacent contacts. For analysis, we included bipolar channels only if both contacts were inside gray matter. Next, any DC offset was removed from each bipolar channel. Thereafter, using the Brainstorm software (*Tadel et al., 2011*), bipolar EEG signals were bandpass filtered in the 3–200 Hz range and notch filtered at 60, 120, and 180 Hz. Bandpass filtering was performed using a Kaiser-window linear phase FIR filter of order 7252 (for a sample rate of 1000 Hz), using the *fir1* MATLAB function. We compensated for filter-induced delay by shifting the filtered sequence, resulting in a zero-delay filter. The notch filtering used an IIR filter of order 6 with a 3dB bandwidth of 1 Hz around the notch frequencies. We used the *filtfilt* function in MATLAB to compensate for group delay. The resulting preprocessed data were employed in all analyses in this paper. When analyzing low-frequency and high-frequency brain connectivity, we bandpass the preprocessed data using the same filter type (Kaiser-window linear phase FIR filter) with different ranges and orders.

## High-frequency ictal networks

The time-varying brain networks were computed in two frequency bands, 80–140 Hz, and 140–200 Hz. Briefly, we first applied the Hilbert transform to compute analytical signals in each frequency band. Dynamic connectivity matrices were calculated using pair-wise lagged-coherence (*Pascual-Marqui, 2007*) between signals in 2.5 s windows with 80% overlaps. Here, the term 'synchrony' is employed interchangeably with brain connectivity in high-frequency. Lagged-coherence removes spurious coherence values caused by volume conduction. As a result, we have time-varying N×*N* networks in two frequency bands for each seizure, in which *N* denotes the number of bipolar channels. Network calculation was performed using Brainstorm (*Tadel et al., 2011*). The connectivity matrices were z-scored with respect to the pre-ictal period and their values were mapped into the interval (0 1) using an exponential transform (*Burns et al., 2014*). Assuming a total of *T* overlapping 2.5 s windows during a seizure, in which *T* depends on the length of the seizure and window parameters, results in a three-dimensional (N×N×T) matrix or network for each frequency band.

## mlEVC and its decomposition

We constructed an $NT \times NT$ super-adjacency (connectivity) matrix $\mathcal{A}$ with coupling effects between layers (*Kivela et al., 2014*). The diagonal N×N blocks were the adjacency matrices at different time points. Off-diagonal terms were identity matrices multiplied by a coupling parameter *c*, in which $c \in \{1, 2, \dots, 10, 15\}$. Matrix $\mathcal{A}$ is irreducible for any *c*>0. The weighted identity blocks represent ordinal interlayer links between nodes corresponding to a particular contact (neighboring time points). One can consider the leading eigenvector of matrix $\mathcal{A}$ ($\varphi_1 \in \mathbb{R}^{NT}$) as the EVC of this super-graph (*Solá et al., 2013*). However, in our work, this vector was focused on a few adjacent layers and did not visually capture the centrality across all layers. This is because of the time-varying nature of ictal networks and the relatively weak coupling across time in our model. Historically, the EVC is defined as a weighted sum of all eigenvectors of the matrix but simplified to the leading eigenvector for matrices

with one clique (**Bonacich, 1972**). In our model of time-varying brain connectivity, this simplification does not hold because of the time-varying nature of connectivity during the seizure that is embodied in the super-adjacency matrix. We, therefore, defined a mlEVC that combines the eigenvectors corresponding to the largest eigenvalues. As discussed in the results section, we chose $T$ eigenvectors irrespective of the coupling parameter. In our observations, the eigenvectors $\varphi_1, \ldots, \varphi_T$ were each restricted to significant values across a few layers only. The Perron-Frobenius theorem asserts the positivity of $\sigma_1$ (the largest eigenvalue) and $\varphi_1$ but the elements of other eigenvectors can be non-positive ($\varphi_i \preccurlyeq 0 \ for \ 2 \leq i \leq T$). Since EVC is a measure of ranking between nodes, we considered the absolute values for all vectors. To extract the mlEVC for matrix $\mathcal{A}$, the $T$ largest eigenvalues were multiplied by their corresponding eigenvectors and the absolute values of the results summed,

$$mlEVC = \left|\sigma_1\varphi_1\right| + \left|\sigma_2\varphi_2\right| + \ldots + \left|\sigma_T\varphi_T\right|$$

This vector was then reshaped to an N×$T$ matrix, representing the variation in each node's centrality over time, which we refer to as the mlEVC. In the case where $c=0$, mlEVC represents the concatenated EVC of the adjacency matrices computed separately for each time point. Elements with the smallest or highest values show isolated or strongly connected instances in time, respectively. Since EVC provides a connectivity ranking among nodes, the mlEVC of each seizure was quantized based on a percentile thresholding ($d$). The top $d/2$-portion of elements was assigned a value of '1,' the bottom $d/2$-portion a value of '−1,' and the remainder a value of '0.' We concatenated the quantized mlEVC matrices into an $N \times T_{tot}$ matrix, in which $T_{tot} = \sum_{s=1}^{I}\sum_{f=1}^{2} T_{sf}$ , where $T_{sf}$ denotes the number of samples in the network for the $s^{th}$ seizure and $f^{th}$ frequency band. This matrix contains the centrality measure for $I$ ictal periods in two frequency bands. The singular value decomposition (SVD) was applied to this matrix to find the left and right singular vectors : $\boldsymbol{u}_i \in \mathbb{R}^N$ and $\boldsymbol{v}_i \in \mathbb{R}^{T_{tot}}$ . The left singular vectors summarize each node's characteristics across all seizures in the context of centrality.

## Unsupervised clustering for EZ identification

To identify candidate contacts for the EZ, weighted consensus clustering (**Kiselev et al., 2017**; **Li and Ding, 2008**) was employed, using combinations of the first four left singular vectors. These vectors were z-scored and used as features in a hierarchical clustering algorithm. This approach clusters the nodes in a bottom-up agglomerative fashion. Based on our hypothesis, the EZ should have a distinctive pattern of connectivity and therefore centrality. Our goal is, therefore, to find a dense target cluster of nodes (X=$1$) whose feature vectors have a significant distance from the centroid of the feature vectors of the other nodes (Y=$0$). As a result, we trace the dendrogram to the last step where the final two groups merge. For a fixed coupling and threshold, $c$ and $d$, respectively, feature vectors were selected from a pool of normalized singular vectors, consisting of the five combinations:

$$\mathscr{K} = \left\{\left\{\boldsymbol{u}_1, \boldsymbol{u}_2\right\}, \left\{\boldsymbol{u}_1, \boldsymbol{u}_3\right\}, \left\{\boldsymbol{u}_2, \boldsymbol{u}_3\right\}, \left\{\boldsymbol{u}_1, \boldsymbol{u}_2, \boldsymbol{u}_3\right\}, \left\{\boldsymbol{u}_1, \boldsymbol{u}_2, \boldsymbol{u}_3, \boldsymbol{u}_4\right\}\right\}$$

For each combination, $\kappa_r \in \mathscr{K}, \ r = \{1, \ldots, 5\}$, the feature vectors of dimensions 2–4 were integrated into the MATLAB hierarchical clustering function *linkage (centroid with Euclidean distance)*. Nodes were divided into two groups before the last linkage. We assigned a binary label '$1$' to nodes in the smaller cluster (presumptive target X) and '0' to other channels (presumptive Y). Since hierarchical clustering is prone to outliers, we went one step back in the dendrogram if the minor cluster consisted of less than 5% of the nodes. In other words, we divided the nodes into three groups and selected the second minor group as the presumptive target.

Ideally, we would like to see a tight target cluster that is well separated. Consequently, one can evaluate a clustering technique by computing the quotient (**Halkidi et al., 2001**),

$$perf\left(r\right) = \frac{sep\left(r\right)}{comp\left(r\right)}$$

where the separation index for the $r^{th}$ combination is defined as the Euclidean distance:

$$sep\left(r\right) = \left\|\bar{\mathcal{Y}}_r - \bar{\mathcal{X}}_r\right\|_2^2$$

where $\bar{\mathcal{X}}_r$ and $\bar{\mathcal{Y}}_r$ are the centroids of target and not-target clusters. Compactness is defined as the product of two measures of intra-cluster distance in the target group,

$$comp\,(r) = \left(\tfrac{1}{M} \sum_i \sum_{j>i} \|\mathbf{x}_{ir} - \mathbf{x}_{jr}\|_2\right) \left(max_i \, \|\mathbf{x}_{ir} - \bar{\mathcal{X}}_r\|_2\right)$$

in which $x_{ir}$ is the feature vector of the $i^{th}$ point in the target cluster and $M$ is the total number of pairs. The first term calculates distances between all nodes and the second term finds the farthest distance of a point from the center of the cluster.

We combined the hierarchical clustering results for all sets of feature vectors. The vector $\mathbf{w} \in \mathbb{R}^N$ is calculated as the probability that each node is a candidate for EZ,

$$\mathbf{w} = \frac{1}{\sum_{r=1}^{n} perf\,(r)} \sum_{r=1}^{n} perf\,(r)\,\mathfrak{l}_r$$

where $\mathfrak{l}_r \in \mathbb{R}^N$ is the hierarchical labeling result for the $r^{th}$ case. The parameter n≤5 is defined as the number of feature vector sets that lead to clustering where $comp\,(r) \neq 0$ which in turn requires a minimum of two contacts in the EZ cluster. Since we need a EZ vs non-EZ label for each node, the vector $\mathbf{w}$ was binarized using a threshold $\theta = \frac{n-1}{n}$ . For the specific case n=1, $\theta = 0.5$. We constructed the binary vector $\mathfrak{l} \in \mathbb{R}^N$ by thresholding $\mathbf{w}$,

$$l_i = \begin{cases} 1 & w_i > \theta \\ 0 & otherwise \end{cases}$$

We repeated the above procedure for a range of parameter values: $c \in \{1, 2, \ldots, 10, 15\}$ and $d \in \{0.1, 0.2, \ldots, 0.8\}$. The final consensus vector $\mathbf{w} \in \mathbb{R}^N$ was then computed to represent the overall chance of a node being in the target (EZ) cluster as

$$\mathbf{w} = \sum_c \sum_d perf\,(c,d)\,\mathbf{l}_{cd}$$

The vector $\mathbf{w}$ is continuous where larger values indicate increasingly likely candidates for the EZ. We applied $k$-means clustering ($k$=3) on vector $\mathbf{w}$ and took the cluster with the largest average value as the final target (EZ) group of contacts. Note that in *Figure 2* and *Figure 2—figure supplement 1* we use a single fixed $c$ and $d$ for illustration purposes.

## Brain states

The seizure evolution is captured by the right singular vectors ($\mathbf{v}_i \in \mathbb{R}^{T_{tot}}$). To preprocess the data, the singular vectors were separated into distinct seizures and frequency bands. For each vector, the MATLAB outlier removal function *hampel* was then used to remove outliers, with 15 neighboring points and three scaled median absolute deviations (1.4826 × MAD). These vectors were then recon-catenated in two frequency bands ($\widetilde{\mathbf{v}}_{if} \in \mathbb{R}^{T_{tot}/2}$) where $T_{tot}$ was defined as two times the length of seizures. We then applied K-means clustering, separately for each possible combination of four drawn from the six vectors corresponding to the first three pre-processed singular vectors in each of the two frequency bands. In other words, selecting from the following set,

$$\mathscr{S} = \left\{\widetilde{\mathbf{v}}_{11}, \widetilde{\mathbf{v}}_{12}, \widetilde{\mathbf{v}}_{21}, \widetilde{\mathbf{v}}_{22}, \widetilde{\mathbf{v}}_{31}, \widetilde{\mathbf{v}}_{32}\right\}$$

We repeated the analysis for different numbers of clusters and used the MATLAB *silhouette* function to determine the optimal number of clusters and best set of vectors. This metric compares the distance of each point with other points in its cluster to the distances to other clusters. The clustering with the highest Silhouette value was selected to construct the transition matrices and find the transitions among brain states (clusters) shown in *Figure 3* and *Figure 3—figure supplement 1*. The final four selected vectors from $\mathscr{S}$ were denoted $\bar{\mathbf{v}}_a$, $\bar{\mathbf{v}}_b$, $\bar{\mathbf{v}}_c$ , and $\bar{\mathbf{v}}_d$ .

## Statistical analysis of functional connectivity in high-frequency

To quantify brain dynamics, time-varying connectivity measures were constructed for each seizure and frequency band, based on the connectivity between brain regions as follows: EZ-nR, RnEZ-nR, and

nR-nR, where EZ, RnEZ, and nR represent respectively predicted EZ (identified using the methodology described above), resection region not in the EZ, and non-resected areas. Each measure was defined by averaging all connectivity values between nodes in the two regions. As a result, we have three synchrony time-series vectors for each seizure/patient/frequency (s/p/f) at 2 samples/s.

We examined these time series over three sub-intervals: pre-ictal (−0.3,−0.1) L, mid-seizure (0.3, 0.5)*L,* and post-ictal (1, 1.2)*L*. For each synchrony measure, we computed the average value of connectivity in these intervals, resulting in nine values for each s/p/f (39 seizures in total and two frequency bands). We employed a robust percentile bootstrap test (*Wilcox, 2011*) with a one-step M-estimator to perform statistical tests for pairwise comparison between and within the three measures at each of the three periods (function *rmmcppb* in WRS2 package (*Wilcox, 2011*) for R). Computed p-values were corrected using Hochberg FDR correction for J=18 comparisons. The results are shown in *Figure 4* and *Figure 4—source data 1* . The EZ-nR connectivity in the mid-seizure was the most remarkable characteristic of the ictal period. Consequently, we used this to explore how the feature changes among patients. For each participant, we averaged the value of EZ-nR among all seizures. We then performed robust regression of these values using bootstrap sampling and the Theil-Sen algorithm (*Wilcox, 2011*) against both patient age and duration of epilepsy as shown in *Figure 5*.

## Low-frequency propagation networks

Data was filtered in the range of 3–50 Hz. Propagation networks were computed using the PLI over a moving window with a 2.5 s length and 90% overlap. Connectivity matrices were computed from normalized PLI values using element-wise z-scoring with respect to the pre-ictal period and their values were mapped into the interval (0 1) using an exponential transform (*Burns et al., 2014*). We used the previously defined measures, EZ-nR, RnEZ-nR, and nR-nR, to explore interactions among brain regions at these lower frequencies. An analogous analysis to the last section was performed except the selected time intervals were pre-ictal (−0.3,−0.1) L, early-seizure (0,0.2)*L,* and pre-termination (0.8,1)*L*. These results are presented in *Figure 6* and *Figure 6—source data 1*.

## Acknowledgements

Research reported in this paper was supported in part by the National Institutes of Health under awards R01NS089212 and R01EB026299.

## Additional information

### Funding

| Funder | Grant reference number | Author |
|---|---|---|
| National Institute of Neurological Disorders and Stroke | R01NS089212 | Dileep R Nair |
| National Institute of Biomedical Imaging and Bioengineering | R01EB026299 | Richard M Leahy |

The funders had no role in study design, data collection and interpretation, or the decision to submit the work for publication.

### Author contributions

Hossein Shahabi, Conceptualization, Data curation, Software, Formal analysis, Validation, Investigation, Visualization, Methodology, Writing – original draft, Writing – review and editing; Dileep R Nair, Conceptualization, Resources, Data curation, Supervision, Funding acquisition, Investigation, Writing – original draft, Project administration, Writing – review and editing; Richard M Leahy, Conceptualization, Resources, Formal analysis, Supervision, Funding acquisition, Validation, Investigation, Methodology, Writing – original draft, Project administration, Writing – review and editing

## Author ORCIDs

Hossein Shahabi (ID) http://orcid.org/0000-0003-3600-8007
Richard M Leahy (ID) http://orcid.org/0000-0002-7278-5471

## Ethics

This retrospective study was approved by the institutional review board at the Cleveland Clinic. Single pulse electrical stimulation induced cortico-cortical evoked potentials are collected as a part of the routine clinical care of patients undergoing SEEG at Cleveland Clinic. The ictal data is also collected during the presurgical SEEG evaluation. The full procedure for participant selection and data recording is described in our previous work (Grinenko et al., 2018).

## Decision letter and Author response

Decision letter https://doi.org/10.7554/eLife.68531.sa1
Author response https://doi.org/10.7554/eLife.68531.sa2

---

## Additional files

### Supplementary files
• MDAR checklist

### Data availability

The results and codes generated during the current study are available in the following repository, https://data.mendeley.com/datasets/t8bvh5m8bp/1.

The following dataset was generated:

| Author(s) | Year | Dataset title | Dataset URL | Database and Identifier |
|---|---|---|---|---|
| Shahabi H, Leahy R, Nair D | 2022 | Brain Connectivity During Seizures | https://doi.org/10.17632/t8bvh5m8bp | Mendeley Data, 10.17632/t8bvh5m8bp |

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
