## [Editor Report]

This valuable work proposes new network-based algorithms for brain seizure characterisation that could improve the effectiveness of existing clinical treatment paradigms. The approach is supported by solid evidence. If validated and compared against existing biomarkers, it could shed light on mechanisms of disease progression. This work will be of interest to clinicians and researchers in epilepsy alike.

---

## [Decision Letter]

**Decision letter after peer review:**

Thank you for submitting your article "Multilayer brain networks can identify the epileptogenic zone and seizure dynamics" for consideration by *eLife*. Your article has been reviewed by 3 peer reviewers, and the evaluation has been overseen by a Reviewing Editor and Michael Frank as the Senior Editor. The following individuals involved in review of your submission have agreed to reveal their identity: Mangor Pedersen (Reviewer #2); Christian Benar (Reviewer #3).

Essential revisions:

1. The prediction of the EZ seems to be completely within-sample. Please provide some indication of cross-validated out-of-sample prediction accuracy.

2. Please evaluate network properties with respect to appropriate null models. An important component of assessing the presence of network structure is to evaluate whether that same structure or nodal importance is not evident in null model where the underlying network structure is randomized. This is key to making sure that clustering results are not just sensitive to random fluctuations in the data. One suggestion is to use a phase-randomization of the voltage trace before computation of HFS to help better evaluate the null distribution of the mlEVC metric. A related reference is: https://pubmed.ncbi.nlm.nih.gov/22343126/

3. The authors may consider that at least one recent study has discussed the challenges and considerations of different recording and electrode sampling techniques on network metrics. And may explain discrepancies in the findings across studies: https://pubmed.ncbi.nlm.nih.gov/32537538/

4. The methods used in this study are complex, and rationale behind the series of steps used to perform the study are not provided. Given that this paper is primarily showcasing a novel method, the authors may want to consider incorporating more methodological details and rationale in-line with the Results.

5. Providing intuition behind abstract mathematical concepts would be helpful. For example (lines 139-141): Help the reader interpret the hierarchical unsupervised clustering applied to the left singular vectors of the concatenated and quantized mlEVC.

6. Line 96: Consider describing the concepts of graph/super-graph, and single/multi-layer in a general sense, first, as this is the first use of these terms in the manuscript.

7. Line 262: Typo, should "EZ-RnRZ" read "EZ-RnEZ".

8. The seizure evolution and state transition plots can be difficult to follow, due to the combination of colors and hues. Please consider a flow field or separate plots per seizure help demonstrate the overlapping and non-overlapping dynamics between seizures?

9. The authors may want to consider incorporating additional details in Table 1, pertaining to: (a) size of the network; (b) number of nodes overlapping with the resected tissue; (c) durations of the seizures; (d) seizure type.

10. One previous study motivated the application of phase and lagged-based connectivity metrics on the analytical amplitude of the broadband signal to avoid computing phase relationships within the asynchronous broadband range of the voltage signal. See: https://pubmed.ncbi.nlm.nih.gov/12631571/. Note that this is distinct from applying phase-based connectivity metrics directly to the high-frequency broadband component of the voltage signal or to high-frequency narrowbands (such as HFOs).

11. If I understand the methodical approach correctly, the authors generate an ordinal graph optimal when using fast intracranial electrophysiological data (500 or 1000Hz). It would be interesting to see a correlation plot between the current eigenvector centrality with an ordinal coupling parameter and cross-correlation of this data at multiple time-lags. In addition to tuning the coupling strength, the current work would be improved to quantify the optimal time interval between matrix 'layers'. It would be useful for the authors to consider/discuss this point in detail.

12. Related to the previous comment, it would be good to head further why the authors choose to use the Phase Lag Index for slow frequency estimation, a measure that does not operate proximately to the zero-lag.

13. A clearer rationale for using a spatiotemporal graph if warranted in this paper. The authors outline and justify the use of multilayer graphs from a methodological point of view, but it would also be good to hear why the authors choose this methodology from a clinical point of view. i.e., what can multilayer metrics tell us about epilepsy that single-layer cannot?

14. Related to the previous point, have the authors thought about comparing eigenvector centrality between multilayer and single-layer graphs, to see whether the results differ between the two?

15. Please provide more details about the filtering procedure, conducted before the Hilbert transform, including the type (and order) of filtering, how the DC offset was accounted for and whether the Hilbert transform in the given frequencies satisfy the Bedrossian's criteria for narrow-band frequency estimation (Xu and Yan, 2006).

16. It would have been nice to see in more detail examples from one or two patients with the actual resection area and the detected networks in volume. Figure 1 would benefit from an intuitive description of the method.

17. P7 It seems to me the term 'primary organization' has to be credited to Talairach and Bancaud. Also, early work on network organization were performed by Wendling and Bartolomei and could be cited (including the desynchronization at seizure onset in Wendling et al. 2003).

18. P8 it is very interesting to see that in a few patients one method fails whereas the other gives good results. This pleads for a complementarity of strategies to cope with inter-patient variability. Can the authors make hypotheses on which signal features/seizure patterns could explain the difference in results across methods and patients?

19. P11 « false positives from the EZ » this really depends on the definition of the EZ. If this is the minimum of cortex to be resected to render the patient seizure free, then the statement may hold. If this is the « primary region of organization of seizures » (definition of Bancaud, given p7), it is possible that not all these regions need to be resected. It would be useful to clarify the two definitions.

20. p17 it would be very interesting to perform a comparison of HF and LF synchrony in predicted the EZ. Moreover, as mentioned below too, part of the HF synchrony can arise from trains of sharp spikes that have energy in all frequencies, which would not be synchrony of high frequency oscillations but an emphasis on the sharp part of ictal spikes. This is not a problem per se, but more an issue of interpretation of the results.

21. Is data processing on monopolar or bipolar montage? Please clarify

22. Is it possible that part of the synchrony arises from filtered spikes? Can the authors provide a (normalized across frequencies) time-frequency representation of a representative seizure in order to assess the relative contribution of oscillations and spikes?

23. P25, eigenvalue centrality. While the rank of the matrix is likely to be T, wouldn't one expect some kind of temporal correlation across points (arising from both the high overlap and some actual temporal persistence of the networks), which would result in an elbow in the eigenvalues after some dimension? If it is not the case, and each time point is independent from the other ones (linear eigenvalue spectrum), what is the added power of multilayer method? Wouldn't a more simple method as summing the degrees across time be equivalent? Please clarify.

24. It would be interesting to visualize the eigenvalues of a representative example in order to measure this. Also, and importantly, it would be useful to compare the multilayer method to a simple mean degree across time (Wilke et al. 2010, van Mierlo et al. 2013, Courtens et al. 2016, Li et al. 2016, Balatskaya et al. 2020).

References:

Kramer, M.A., Eden, U.T., Kolaczyk, E.D., Zepeda, R., Eskandar, E.N., Cash, S.S., 2010. Journal of Neuroscience. 30, 10076-10085.

Schindler, K.A., Bialonski, S., Horstmann, M.-T., Elger, C.E., Lehnertz, K., 2008. Chaos 18, 033119.

Xu, Y and Yan, D, 2006. Proceedings of the American Mathematical Society 134, 2719-2728

*Reviewer #1:*

In this study, Shahabi and colleagues develop a new computational algorithm based on graph theory to study changes in brain network connections during drug-resistant, epileptic seizures. The structure and organization of these network connections have been the subject of many studies over the past two decades with the principal purpose of mapping seizures and identifying targets for resective surgery. A novel analysis of distributed brain network interactions between the local epileptogenic zone and remote, healthy brain areas outside of this zone is facilitated by invasive stereo-electroencephalographic (sEEG) recordings, which are a key asset to the present study. The authors construct time-dependent functional brain networks based on inter-areal synchrony in the high-frequency band and combine a new graph theory metric called multilayer eigenvector centrality (mlEVC) to quantify the changing pattern of connectivity amongst brain areas within and outside the epileptogenic zone. Using their innovative technique, the authors recapitulate a widely reported result that seizure progression invokes a robust alteration in network organization in which connectivity between the epileptogenic zone and healthy brain areas desynchronize. However, a key contribution of the present study is that the network nodes in the epileptogenic zone whose subsequent surgical resection led to seizure freedom could be predicted using unsupervised machine learning. These findings suggest that nodes involved in desynchronization during seizures may serve as putative surgical targets for epilepsy treatment. The authors also demonstrate that if patients are left untreated, then this abnormal desynchronization process during seizures intensifies with age and duration of epilepsy. This is a compelling scientific advancement in the field that begins to tackle the long-standing question of whether seizures beget seizures.

Generally, the study and analysis are presented as exploratory and proof-of-principle. A major strength of this study is the development of a new methodology to describe complex properties of seizures. However, I have concerns regarding possible overfitting of the data as no cross-validation nor testing was performed. Key claims could be better motivated with concrete hypotheses that are contextualized by prior work. My specific comments are detailed below:

1. Key questions posed in the study are provided and/or stated in the introduction but in many cases the importance or relevance in the broader context are not fully developed. Examples below:

a. Lines 50-52: "It has also been suggested that ictal periods can be delineated by a steady series of states(Burns et al., 2014), although whether this is true in all patients remains controversial."

b. Lines 52-55: "It remains unclear how the degree of desynchronization is correlated with physiological parameters such as age and duration of epilepsy(Van Diessen et 55 al., 2013a)"

2. Much of the introduction focuses on high frequency oscillations and pros/cons of their utility in localizing seizure foci. The authors present ictal high frequency synchronization as a phenomenon of interest, but to the general audience the differences between HFO and HFS are not concretely provided. The rationale for studying ictal HFS over HFOs is also not provided. Are these the same phenomenon? Are they distinct? Why do authors deem HFS analysis important for this study?

a. Lines 75-77: "While HFOs have been employed in analyzing functional(Schindler et al., 2010) and propagation(González Otárula et al., 2019) networks, the spatiotemporal dynamics of ictal high frequency synchronization (HFS) at macroscopic scales remain largely unknown."

b. Lines 238-240: A later point conflates HFOs and HFS as the same phenom: "Early-onset and late ictal HFOs have been considered biomarkers for seizure onset zone identification (Weiss et al., 2013), with the latter found to be a more reliable metric (Modur et al., 2011). Our EZ localization technique considers both features."

3. The interpretation of connectivity in the high-frequency broadband is unclear in the context of previous studies that demonstrate that the broadband activity is due to asynchronous, non-oscillatory neural firing (see references below). Consequently, it is unclear how one should interpret the notion of "synchronization" in the broad 80-200 Hz frequency range.

a. https://pubmed.ncbi.nlm.nih.gov/23283342/

b. https://pubmed.ncbi.nlm.nih.gov/29167419/

4. Was a null network model employed to ascertain whether the clustering procedure and the mlEVC metric identified target epileptogenic zone areas more reliably than chance?

5. What was the trade-off in the sensitivity and specificity of the EZ prediction algorithm? A receiver-operator curve analysis would help here.

6. One concern is that the analysis corresponding to the prediction of the epileptogenic zone is performed at the population level – aggregating nodes across all patients (line 160). What was the mean and standard error of the performance across patients?

7. The comparison between the present method and Fingerprinting method is of great value here. However, a clear conclusion regarding the comparison is not provided because the two algorithms provided results that only partially overlapped. How do these results compare to more conventional measures of the epileptogenic zone such as HFOs or spikes derived from the same dataset?

8. Results regarding the occurrence of different states of connectivity during the ictal period (lines 177-180) are certainly very interesting, but it is not clear how this finding advances previous studies:

a. https://pubmed.ncbi.nlm.nih.gov/20668192/

b. https://pubmed.ncbi.nlm.nih.gov/23366973/

9. A related worry regarding statistical power of the study is reflected in the stereotypy analysis (lines 192-195). To what extent is the stereotypy / lack of stereotypy in the network a function of the dimensionality of the feature space and overfitting of the data? Specifically, as the number of features / complexity of the model increases, are you more likely to find that each seizure event network is different from the others? A cross-validation and out-of-sample prediction approach would help mitigate these concerns.

10. The finding that stereotypy does not necessarily occur at high-frequencies is very interesting (lines 206-209), but the claim is not supported by any statistical testing. Furthermore, the explanation provided "the brain experiences divergent topologies, which might be the result of dissimilar EEG recordings" (line 209) is rhetorical as the topologies are based on features derived from the EEG recordings.

11. Why was the mid-seizure period used to normalize network measures for age/epilepsy duration-related analysis? How is this baseline period more advantageous than a pre-seizure baseline?

12. Were any other connection groups tested as being predictive of age or duration of epilepsy? In particular, is it possible that prolonged epilepsy might reorganize areas that are purely outside the EZ? Was there any relationship between the size of the EZ and age/duration?

13. Can the authors be sure that the emergence in low-frequency connectivity is not related to an increase in the rate or amplitude of ictal spiking? Are they independent phenomena?

14. How are the phases of the seizure (pre-termination, termination, early-seizure) defined? Are they based on the state analysis or on expert ratings of the seizures? A clear definition of these periods would be helpful.

15. Specific steps and choice of parameters in the unsupervised clustering pipeline described in the Methods are not justified. Such as

a. Usage of just the first four singular vectors

b. Limited combinations of the singular vectors

c. Determination of the rank T for the A matrix

d. How the mlEVC approach compares to the traditional EVC approach similar to that studies in the Burns et al. (2014).

*Reviewer #2:*

The authors combines intracranial EEG data with multilayer network modelling to delineate the seizure onset zone and spatiotemporal network activity during seizures. The major strength of this study is that the networks in this study incorporates spatial and temporal connectivity during seizures, and the relative importance of different intracranial electrodes. This is a novel and interesting approach that is likely to have an impact in the field. Below I discuss two topics (near zero-lag connectivity and previous research in the field) that is worth bearing in mind regarding the current manuscript.

- The authors generate an ordinal graph (connectivity between two adjacent time-points) to quantify spatiotemporal connectivity from intracranial electrophysiological data with a high sampling rate. An ordinal graph computes the connectivity between two adjacent time points, which may represent synchronicity close to a zero-lag in this data. It is worth bearing in mind graphs with connectivity data where time-points that are further apart, and how this may reflect neuronal connectivity at different time-scales.

- The current set of findings are interesting and they show good clinical accuracy at predicting the seizure onset zone. Spatio-temporal connectivity during seizures in this study also align with previous graph theoretical research using intracranial electrophysiological with single-layer graphs. Previous studies show distinct time-varying peri-seizure changes in the clustering coefficient and path length metrics. During seizure initiation and propagation, there is an increase of clustering coefficient and shortest path length, resembling a regularized, or isolated, network topology (Kramer et al., 2010; Schindler et al., 2008). In a network that is regularized, local clusters of neurons may isolate themselves from the rest of the network. It would be good to evaluate the current findings in the context of previous graph theoretical research using intracranial electrophysiological data in epilepsy.

*Reviewer #3:*

This manuscript introduces a new method for identifying the epileptogenic zone from intracerebral EEG data, based on graph measures in high frequency bands, multilayer graph analysis and clustering.

Graph measures and multivariate methods are promising tools in the electrophysiological characterization of the epileptogenic zone; the strategy proposed here falls within this timely topic. The strength of the approach is to be fully automatic, and to rely on multivariate measures that can capture the overall structure of the data better than the usual monovariate measures. The method identified electrodes within the resected area in 88% of the patients, with only few contacts detected outside of the resection.

It could be interesting to also test the method versus the clinician EZ, as the resected area may be larger this would be consistent with the fact that the 'target' area found by clustering is much smaller that the resected contacts. This would be a further probe of the sensitivity of the method. Another interesting measure would be to probe whether the proportion of contacts outside the resected area increase in non seizure-free patients. Yet another interesting test would be to see whether the multivariate method outperforms more classical measures such as graph strength. In other words, it what does the multilayer approach add to the single layer measures?

[Editors' note: further revisions were suggested prior to acceptance, as described below.]

Thank you for resubmitting your work entitled "Multilayer brain networks can identify the epileptogenic zone and seizure dynamics" for further consideration by *eLife*. Your revised article has been evaluated by Michael Frank (Senior Editor) and a Reviewing Editor.

The manuscript has been improved but there are some remaining issues that need to be addressed, as outlined below. Please address these comments comprehensively in your response.

1. In their response and amendments to the main text, the authors claim that their method does not require cross-validation and testing over independent datasets since the approach does not involve training the model parameters across patients and does not access information regarding the resection zone. While it is true that the method is applied to each patient, separately, the methodological choices including, but not limited to, functional connectivity metrics, frequency band selection, percentile threshold selection, still effectively "train" the general algorithm and may be highly specific to the cohort studied here. This issue presents itself as a driver of inter-patient variability, as demonstrated by the conflicting findings between the mlEVC method and Fingerprint method. While this limitation detracts from the methodological advances put forth by this study, but some caution in making the claim that cross-validation/testing is not necessary is warranted. Specifically, the manuscript should include a rationale for why cross-validation was not applied in this context, coupled with an acknowledgement that such an approach in future will be important to verify the current results.

2. The relationship between the coupling parameter and network time-scale should be further elaborated, given the importance of this parameter to the authors' claim that patients may have different rates of network change. Please present data examining how sensitive the findings of the EZ are to the choice of this parameter. Please also provide examples of how different coupling parameters would yield more insight into the different time-scales of network change

3. The findings comparing a single-layer model to the multi-layer model should be explicitly quantified to concretely justify the claim in the Discussion section that single-layer models yielded poorer EZ identification than multi-layer models. How much more accurate was the multi-layer model to the single-layer model?

4. More attention should be paid in the text to the risk of contamination of high frequency synchrony by two processes: (1) Filtering of spikes; and (2) Harmonics of non-sinusoidal oscillations. These topics are addressed in the following study, which could be cited in support of the discussion.

Pitfalls of high-pass filtering for detecting epileptic oscillations: a technical note on "false" ripples. Bénar CG, Chauvière L, Bartolomei F, Wendling F. Clin Neurophysiol. 2010 Mar;121(3):301-10

5. The time frequency figures should be accompanied by a presentation of the signal time course in order to fully appreciate the presence/absence of spikes. While there indeed seems to be spikes in the preictal period (vertical lines) , they present much less energy than the high frequency activity during the seizure. This large increase is quite blurred and indistinct and it is not clear what this actually represents. The original signal time course would help understand which seizure pattern this corresponds to.A consideration of the potential effect of harmonics would also be helpful.

---

## [Author Response]

Essential revisions:1. The prediction of the EZ seems to be completely within-sample. Please provide some indication of cross-validated out-of-sample prediction accuracy.

We were inspired by (Burns et al., 2014) while working on our hypothesis. They state, based on exploratory observations, that:

“We found that, in patients with positive surgical outcomes (i.e., patients who were seizure-free after surgery; Table 1), the focal areas were the least connected in the network (least important) shortly after the onset time of the seizure and we denoted this state as the “isolated focus” (IF) state. We also found that, in many of these patients, the focal areas became most connected in the network in a brain state that occurred during the middle or end of the seizures, and we defined this state as the “connected focus” (CF) state.”

They used the clinical resection information to investigate the dynamics of eigenvector centrality in the seizure onset zone. They observed specific patterns of centrality among different groups of patients. However, (Burns et al., 2014) did not introduce an algorithm that directly predicts the seizure onset zone. We believed that there is a gap in this part of literature and an automated algorithm could be developed to help predict the epileptogenic zone (EZ) from ictal network measures separately in each subject.

We therefore proposed an unsupervised framework that aims to identify the epileptogenic zone without any knowledge about the actual resection volume. Based on the observations in (Burns et al., 2014) and other studies, we hypothesized that the EZ should have a unique pattern of brain connectivity. Our algorithm for EZ identification is the same for all patients and does not involve training using resection labels. Since the algorithm was not trained across subjects, there is no need for a separate out-of-sample validation.

The steps employed for unsupervised clustering are common in the literature. For example, thresholding and quantization have been widely used in different brain network studies. For selecting a proper coupling parameter in our multilayer model, we followed the guidelines in (Betzel and Bassett, 2017).

“Without strong evidence to select one weighting scheme over another, interlayer links are usually assigned the same value, ω, that is sometimes varied over a narrow range. Ideally, there would be a principled, data-driven approach for selecting this value.”

Considering neural mechanisms with different timescales, we hypothesized that the proper interlayer parameter might be different among patients. As a result, in our data-driven approach, we first defined a performance function (Halkidi, 2001) for the unsupervised clustering algorithm *linkage* in MATLAB. As we describe in the method section, the performance function was defined based on our hypothesis that nodes inside the “presumptive EZ” should have a similar pattern of brain connectivity. For the final analysis, we combined the results of performance function over the full range of values for thresholding and interlayer coupling parameter, in an identical manner for each subject. These ranges were held constant among all patients.

To clarify this point in the paper, we added the following to the Results section:

“Cross-validation and testing over a separate independent set were not required for our EZ identification technique since the unsupervised approach was applied independently to each subject without training within or across subjects. Furthermore, resection labels were not employed as part of the prediction algorithm. The same performance function and ranges of coupling and thresholding values were used for all subjects.”

In addition to examining the accuracy of our method on 16 patients, we also performed several other analyses to evaluate validity of our technique, which are described later in this response, with associate modifications in the manuscript. Finally, we note that in the initial version of this manuscript, we used “classification” or “classify” interchangeably with “grouping” or “clustering”. We have clarified the text to make it clearer that all clustering was performed in an unsupervised manner.

2. Please evaluate network properties with respect to appropriate null models. An important component of assessing the presence of network structure is to evaluate whether that same structure or nodal importance is not evident in null model where the underlying network structure is randomized. This is key to making sure that clustering results are not just sensitive to random fluctuations in the data. One suggestion is to use a phase-randomization of the voltage trace before computation of HFS to help better evaluate the null distribution of the mlEVC metric. A related reference is: https://pubmed.ncbi.nlm.nih.gov/22343126/

We consulted the suggested reference (Zalesky et al., 2012) and a related paper for generating surrogate data by time series randomization using Fourier transform and phase scrambling (Prichard and Theiler, 1994). The latter article was employed to create surrogate data for several patients, using their SEEG signals during seizures. The mlEVC metric for phase-randomized signals was calculated with several coupling parameters. We compared results of original times series and surrogate data. Results are presented in new Figure 2—figure supplement 2 – see added text below. The left column portrays the mlEVC for the original data and the right column depicts the same metric when the phase scrambled data was used. We sorted nodes such that those inside the resected volume are on the top of the figures. We observe a clear distinction between the two scenarios, in which the mlEVC of the original time series displays characteristic patterns both in resected and non-resected regions that are not apparent in the scrambled data.

We added this paragraph to the results:

“To explore the validity of our approach, we constructed a null model by also computing the mlEVC from phase-randomized SEEG signals (Prichard and Theiler, 1994). In figure 2—figure supplement 2, we show typical results of the mlEVC measures calculated from the original time-series and phase-randomized data for a single subject. These result show significant differences between original and randomized data in which the characteristic patterns of brain connectivity both in resected and non-resected areas are lost when signals are phase-randomized. We performed this analysis for different patients and seizures with similar findings.”

3. The authors may consider that at least one recent study has discussed the challenges and considerations of different recording and electrode sampling techniques on network metrics. And may explain discrepancies in the findings across studies: https://pubmed.ncbi.nlm.nih.gov/32537538/

In the introduction and discussion, we note possible sources of discrepancies among the results of studies on ictal connectivity. In particular, based on the reviewers’ suggestion we added the following:

“Recent research has investigated the effect of incomplete electrode sampling on network metrics (Conrad et al., 2020), in which eigenvector centrality showed the highest reliability in comparison to other network metrics, when randomly removing network nodes. However, they also showed that confidence intervals in network measures can vary significantly between patients. For this reason, care should be taken when interpreting and comparing different studies.”

4. The methods used in this study are complex, and rationale behind the series of steps used to perform the study are not provided. Given that this paper is primarily showcasing a novel method, the authors may want to consider incorporating more methodological details and rationale in-line with the Results.5. Providing intuition behind abstract mathematical concepts would be helpful. For example (lines 139-141): Help the reader interpret the hierarchical unsupervised clustering applied to the left singular vectors of the concatenated and quantized mlEVC.6. Line 96: Consider describing the concepts of graph/super-graph, and single/multi-layer in a general sense, first, as this is the first use of these terms in the manuscript.

The reviewers suggested adding more description regarding the methodology in the results and introduction section of this paper. Before explaining our changes in detail, we note that the proposed methodology in this paper is dominantly focused on EZ identification. Following the *eLife* format, we incorporated the detailed algorithm description at the end of the paper. In Response 1 above, we elaborated how these steps were chosen based on published studies and common practice in the current literature. To improve the readability of the manuscript, we made the following modifications:

In the introduction, we introduce mathematical concepts including graphs/super-graphs/multilayer.

“Graph analysis provides a mathematical framework for quantification of brain connectivity (Bullmore, 2009). Brain networks may be represented as a graph, G=(V, E),­ in which nodes (V) characterize anatomical regions or electrodes, and edges (E) reflect structural or functional connections among them. Traditionally, brain connectivity in each time sample (layer) of dynamic networks has been evaluated independently, via single graph analysis. However, multilayer analysis allows us to model the entire data with a single super-graph, in which individual graphs for each time-sample are linked. Assuming there are *T* single graphs (time-samples) with *N* nodes in each, the super-graph has *NT* nodes.”

In the Results section, we added references to the eigenvector centrality (EVC) and its formulation. EVC usage in ictal studies and its limitation is also discussed. We then describe why and how a new measure (mlEVC) has been defined based on our multilayer model:

“As discussed in the introduction, we aimed to model ictal brain connectivity by multilayer networks. Additionally, we were interested in quantifying network dynamics using centrality metrics (Rubinov and Sporns, 2010). Among those measures, eigenvector centrality, a rank-based metric which assesses the importance of each node in the network (Bonacich, 1972), has been employed in studying seizures (Burns et al., 2014). Mathematically, the leading eigenvector of the adjacency (connectivity) matrix has been assigned as the eigenvector centrality of the graph when there is one clique (component) in the matrix (Bonacich, 1972). However, in our multilayer model of time-varying brain connectivity, a single vector cannot explain the complex structure of the spatiotemporal networks. Therefore, we introduced a new measure called multilayer eigenvector centrality (mlEVC) that incorporates the top T eigenvectors of the adjacency matrix of the super-graph (see Methods). This allows us to evaluate patterns of nodal centrality and identify regions with similar connectivity characteristics to the rest of the graph. Further, because mlEVC is a function of the interlayer coupling parameter (*c*), we can explore neural processes at different timescales by varying *c*.”

In the results, a new section entitled “Algorithm for predicting EZ using mlEVC” was inserted to explain our algorithm, the rationale behind each step, and the way it was executed:

“We hypothesize that the EZ can be identified as the set of nodes in the graph that exhibit a characteristic and distinct pattern of connectivity to other areas during the seizure. To explore this question, we first quantized the mlEVC of each seizure into three levels based on percentile thresholding (*d*). The top *d/2*-portion of elements were assigned a value of ‘1’, the bottom *d/2*-portion a value of ‘-1’, and the remainder a value of ‘0’. For each subject, quantized measures of mlEVC for two high-frequency bands and all seizures were concatenated to a single matrix with dimension *N* by *T_tot_* (twice the total number of sample points) (see Methods). We applied the singular value decomposition (SVD) to the concatenated matrix and used the left singular vectors (ui∈RN) to identify nodes (SEEG channels) with similar features. As an illustration, Figure 2c depicts u1, u2, and u3 for patient 17.

Next, we applied an unsupervised clustering algorithm using the ui vectors to detect a target cluster that represents the EZ. Following our initial hypothesis, the target cluster should portray a dense and distinctive set of nodes in the feature space with a significant distance from nodes in the non-target group (Figure 2c). We describe the clustering algorithm in detail in the method section. Briefly, we designed a data-driven framework to cluster nodes into two groups using an agglomerative hierarchical clustering technique (function *linkage* in MATLAB). This process was performed for different combinations of ui as features of the clustering algorithm, and a range of values for *c* and *d*, resulting in 440 clustering runs. We weighted each run using a performance function (Halkidi, 2001) that examines the tightness of the target cluster and its separation from other nodes. Finally, using the weighted sum of performance metric for all runs, SEEG contacts were divided into two groups; target and non-target.”

In the methods, we added the following description:

“Historically, the EVC is defined as a weighted sum of all eigenvectors of the matrix, but simplified to the leading eigenvector for matrices with one clique (Bonacich, 1972). In our model of time-varying brain connectivity, this simplification does not hold because of the time-varying nature of connectivity during the seizure that is embodied in the super-adjacency matrix.”

7. Line 262: Typo, should "EZ-RnRZ" read "EZ-RnEZ".

Thanks for pointing this out. Corrected.

8. The seizure evolution and state transition plots can be difficult to follow, due to the combination of colors and hues. Please consider a flow field or separate plots per seizure help demonstrate the overlapping and non-overlapping dynamics between seizures?

We updated the state transition plots based on your recommendation. Please check figure 3d and figure 3—figure supplement 1.

9. The authors may want to consider incorporating additional details in Table 1, pertaining to: (a) size of the network; (b) number of nodes overlapping with the resected tissue; (c) durations of the seizures; (d) seizure type.

We updated Table 1 with the requested information for parts a, b, and c. Unfortunately, we do not have the seizure type in our database.

10. One previous study motivated the application of phase and lagged-based connectivity metrics on the analytical amplitude of the broadband signal to avoid computing phase relationships within the asynchronous broadband range of the voltage signal. See: https://pubmed.ncbi.nlm.nih.gov/12631571/. Note that this is distinct from applying phase-based connectivity metrics directly to the high-frequency broadband component of the voltage signal or to high-frequency narrowbands (such as HFOs).

We understand the reviewers’ questions about the terminology, relation of HFOs and high-frequency synchronization and applying phase-based connectivity metric on broad-band signals.

To clarify the terminology and justify the methods and approach used in our analysis, we first added the following description to the introduction.

“The term HFOs has been attributed to brain activity, with multiple possible physiological and pathological neural mechanisms, between 80-500 Hz (Jacobs et al., 2012) or 30-600 Hz (Engel and da Silva, 2012). This includes high-γ neural activities (80-200 Hz) (Ray and Maunsell, 2011) and broad-band high frequency (Arnulfo et al., 2020).”

Although some epilepsy studies have investigated HFOs with specific patterns (e.g. ripples and fast ripples in the time-domain), the broad definition of HFOs remains unchanged. Our analysis of brain connectivity uses the standard HFO definition and is computed using a synchrony metric.

Regarding use of phase-based measures on broad-band signals we added references to several studies that employ them (see Results section):

“Several studies have applied phase-based connectivity metrics on broad-band high-frequency oscillations. For instance, (Zweiphenning et al., 2016) computed phase lag index (PLI) in two high-frequency bands: ripple (80-250 Hz) and fast ripple (250–500 Hz). Another study, used PLI to compute brain connectivity in broadband 80-250 Hz (Nissen et al., 2016). Last, (Burns et al., 2014) used coherence to investigate ictal networks in γ (25-90 Hz) band.”

We also note, as shown in (Aydore et al., 2013) for intracranial recordings, phase-based measures provide essentially the same information as coherence in most cases.

Importantly, we first divided the 80-200 Hz into two narrower frequency ranges (80-140 Hz, and 140-200 Hz) and then computed the connectivity matrices based on the lagged-coherence metric for each frequency range separately. The mlEVC measure was computed for each frequency range separately and the results then concatenated.

11. If I understand the methodical approach correctly, the authors generate an ordinal graph optimal when using fast intracranial electrophysiological data (500 or 1000Hz). It would be interesting to see a correlation plot between the current eigenvector centrality with an ordinal coupling parameter and cross-correlation of this data at multiple time-lags. In addition to tuning the coupling strength, the current work would be improved to quantify the optimal time interval between matrix 'layers'. It would be useful for the authors to consider/discuss this point in detail.13. A clearer rationale for using a spatiotemporal graph if warranted in this paper. The authors outline and justify the use of multilayer graphs from a methodological point of view, but it would also be good to hear why the authors choose this methodology from a clinical point of view. i.e., what can multilayer metrics tell us about epilepsy that single-layer cannot?14. Related to the previous point, have the authors thought about comparing eigenvector centrality between multilayer and single-layer graphs, to see whether the results differ between the two?

We respond to these three comments together since they are highly relevant. In summary the reviewers requested (a) A discussion about the optimal time interval between different layers of the network, (b) Comparison of results between single-layer and multilayer networks, and (c) A rationale for employing a multilayer approach in epilepsy specifically during seizures.

The common approach to analyzing time-varying brain connectivity is using sliding windows with a constant window size and overlap between windows across time. If the interval between windows is small, instantaneous fluctuations will dominate processes with slower timescales. In the case of large intervals between windows, fast dynamics might be discarded. As a result, choosing an optimal interval, is a challenging issue. Applying the same interval for all participants will exacerbate the problem since seizure dynamics are highly variable across subjects.

In our multilayer model of brain networks, we use a 500ms time interval between layers to allow us to capture fast seizure dynamics. By increasing the coupling parameter among adjacent layers, we can incorporate processes with slower time scales. Using this approach, we only need to tune the coupling parameter to extract desired spatiotemporal processes. So, rather than finding an optimal time interval for the multi-layer network, we instead focus on finding an appropriate coupling parameter. We use a data-driven approach to evaluate the cost function for EZ identification over a range of values for the coupling parameter (See Response #1).

We added the following explanation to the discussion.

“By adjusting the coupling between layers, we can overcome the current challenges for selecting an appropriate time interval between different samples of a temporal network. Here, we assumed a relatively short time interval between layers (500ms) to have a high temporal resolution. By increasing the coupling parameter between adjacent layers, we can investigate processes with slower timescales.”

To assert the importance of a multilayer approach when compared to single-layer, we inserted this paragraph into the results.

“Furthermore, we examined the effect of multilayer modeling and adjusting the coupling parameter on EZ prediction and super-graph structure. In figure 2—figure supplement 1, we display the left singular vectors of mlEVC. For illustration purposes, we showed projections with respect to the singular vectors for the parameter set (*c,d*) for which the prediction of EZ (binary labeling) was closest to the assumed ground truth (resection information). We observed that the optimal coupling value is different between patients, which supports our primary rationale for choosing a multilayer framework with an adaptive interlayer coupling parameter. In our initial analysis, we ran the algorithm for the single-layer case and EZ identification results were poor.”

After discussing the methodological reasons for using a multilayer framework, we also explained the neurophysiological rationales. The details as appeared below were added to the introduction:

Additionally, there exist several neurological rationales for employing a multilayer approach in seizure analysis. First, the concept of dynamic network reconfiguration has been studied in brain networks (Bassett et al., 2011; Braun et al., 2015). Previous research has shown state transitions during seizures, either through brain connectivity analysis (Burns et al., 2014) or microelectrode recordings (Smith et al., 2016). Multilayer networks have the capability to delineate these transitions and identify network reconfiguration (Mucha et al., 2010). Second, electrophysiological signals are highly non-stationary during ictal periods. As a result, traditional analysis of time-varying networks based on isolated graphs would be affected by instantaneous fluctuations rather than the underlying spatiotemporal networks. Third, seizure propagation is one of the key elements of ictal activity. Recent work has indicated the importance of multilayer modeling of complex systems when encountering spreading processes (De Domenico et al., 2016).

12. Related to the previous comment, it would be good to head further why the authors choose to use the Phase Lag Index for slow frequency estimation, a measure that does not operate proximately to the zero-lag.

In addition to the brief justification for choosing this measure in the initial version of the manuscript, we added the following description to the result section:

“The PLI is recognized for its superiority to measures such as phase locking value (PLV). By discarding interactions with a phase difference of zero, the PLI quantifies phase coupling between two signals while excluding volume conduction as a confounding factor. Compared to other functional connectivity metrics, PLI is therefore less susceptible to common sources. Since a major part of low-frequency (<50Hz) brain activity during seizures is derived from ictal discharges, which exhibit distance-dependent delays (Smith et al., 2016), true neurological zero-lag connectivity is highly unlikely in this scenario. Because of its robustness to volume conduction, PLI has been widely used in studies of low-frequency (<50 Hz) brain connectivity in epilepsy (Nissen et al., 2016; Schevon et al., 2012; Van Diessen et al., 2013; Zweiphenning et al., 2016).”

15. Please provide more details about the filtering procedure, conducted before the Hilbert transform, including the type (and order) of filtering, how the DC offset was accounted for and whether the Hilbert transform in the given frequencies satisfy the Bedrossian's criteria for narrow-band frequency estimation (Xu and Yan, 2006).21. Is data processing on monopolar or bipolar montage? Please clarify.

Thanks for bringing up these important issues. Regarding the signals montage and preprocessing, we added the following paragraph to the “Patients and recordings” subsection in the methods:

“We preprocessed SEEG signals before further analyses. First, we constructed bipolar channels by subtracting unipolar signals recorded from each pair of adjacent contacts. For analysis we included bipolar channels only if both contacts were inside gray matter. Next, any DC offset was removed from each bipolar channel. Thereafter, using the Brainstorm software (Tadel et al., 2011), bipolar EEG signals were bandpass filtered in the 3-200 Hz range and notch filtered at 60, 120, and 180 Hz. Bandpass filtering was performed using a Kaiser-window linear phase FIR filter of order 7,252 (for a sample rate of 1000 Hz), using the *fir1* MATLAB function. We compensated for filter-induced delay by shifting the filtered sequence, resulting in a zero-delay filter. The notch filtering used an IIR filter of order 6 with a bandstop of 1 Hz in notch frequencies. We used the *filtfilt* function in MATLAB to compensate for group delay. The resulting preprocessed data were employed in all analyses in this paper. When analyzing low-frequency and high-frequency brain connectivity, we bandpassed the preprocessed data using the same filter type (Kaiser-window linear phase FIR filter) with different ranges and orders.”

In Author response image 1 we show the frequency response of the bandpass and notch filters and their characteristics. The low transition band (2.5-3 Hz) is also magnified for a better visualization. In the early version of this manuscript, we listed (2-200 Hz) as the range of the bandpass filter. This has been corrected to (3-200 Hz) in the revised manuscript. The data were filtered over a sufficiently long interval either-side of the seizure window that transient effects of filters was not a matter of concern. Author response image 2 demonstrates the frequency response for the notch filter.

As stated in response #10, we bandpassed preprocessed signals in two narrower frequency ranges, (80-140 Hz) and (140-200 Hz), before applying Hilbert transform, satisfying the Bedrosian identity (Xu and Yan, 2006).

**Author response image 1. sa2fig1:** Magnitude and impulse response of the FIR bandpass preprocessing filter. The upper figure shows an expanded view of the low-frequency response of the filter.

**Author response image 2. sa2fig2:** Magnitude and impulse response of the notch filter. The upper figure shows an expanded view of one of the notch bands.

16. It would have been nice to see in more detail examples from one or two patients with the actual resection area and the detected networks in volume. Figure 1 would benefit from an intuitive description of the method.

We appreciate your suggestion. Unfortunately, we did not have access to MRIs with co-registered electrodes and resected volumes while preparing this response letter.

17. P7 It seems to me the term 'primary organization' has to be credited to Talairach and Bancaud. Also, early work on network organization were performed by Wendling and Bartolomei and could be cited (including the desynchronization at seizure onset in Wendling et al. 2003).

Thanks for the suggestions. We now cite the following publications in the article.

Talairach J, Bancaud J. 1966. Lesion, “irritative” zone and epileptogenic focus. *Confin Neurol* 27:91–94. doi:10.1159/000103937

Wendling F, Bartolomei F, Bellanger JJ, Bourien J, Chauvel P. 2003. Epileptic fast intracerebral EEG activity: Evidence for spatial decorrelation at seizure onset. *Brain* 126:1449–1459. doi:10.1093/brain/awg144

18. P8 it is very interesting to see that in a few patients one method fails whereas the other gives good results. This pleads for a complementarity of strategies to cope with inter-patient variability. Can the authors make hypotheses on which signal features/seizure patterns could explain the difference in results across methods and patients?

The result section mentioned that insufficient data (short seizures) might be one reason for mlEVC's failure. Here, we added another analysis to look for signal features. For patient 8, mlEVC could not identify any nodes inside the resected area as EZ, while Fingerprint predicted two nodes as EZ (one false positive). We looked at the time-frequency spectrum and mlEVC for one seizure of this subject, which is illustrated in Author response image 3. In the upper figure, we can see the main features of the Fingerprint algorithm (suppression, fast activity, and pre-ictal spikes) in the predicted nodes. However, we do not observe any significant HFOs for those nodes (see also our response to comments #20 & #22 regarding HFOs). As a result, the mlEVC does not have a distinctive pattern in any nodes in the resected region (Author response image 3, lower). Lack of dominant HFOs in the EZ can be another factor attributes to mlEVC's failure.

On the other hand, Fingerprint can fail for different reasons. First and foremost, its features might not be present for some seizures or patients. Second, since Fingerprint uses a trained model for predicting the EZ in the new patient, the new features can be quite different from the base model, or the detection process falls short of the presumed threshold for EZ identification.

Seizure types and locations can also affect the results in both methods, but it is hard to draw conclusions with a small number of patients in each cohort. Additional studies on a larger dataset are needed to further investigate these issues, which is beyond the scope of the current paper.

**Author response image 3. sa2fig3:** Time-frequency spectra and mlEVC for one seizure of patient 8.

19. P11 « false positives from the EZ » this really depends on the definition of the EZ. If this is the minimum of cortex to be resected to render the patient seizure free, then the statement may hold. If this is the « primary region of organization of seizures » (definition of Bancaud, given p7), it is possible that not all these regions need to be resected. It would be useful to clarify the two definitions.

We use the first definition since with this, we can definitively state that the EZ was not resected in those patients who are not seizure free. The EZ is discussed in the Introduction as follows:

“In using SEEG to identify the epileptogenic zone (EZ) we attempt to take into account the earliest ictal EEG change based on an anatomoelectroclinical analysis (Kahane et al., 2006). This concept incorporates both the anatomic region that initiates the epileptic discharge as well as the “primary organization” (Talairach and Bancaud, 1966) that leads to the manifestation of the clinical seizure itself (Wyllie et al., 2015). The gold standard method of confirming EZ localization is based on whether seizure freedom has been achieved by resection or ablation. However, the actual ground truth for EZ location, in the sense of the minimal resection volume needed to achieve seizure freedom, is unknown since in many cases the resection volumes may extend well beyond the EZ.”

20. p17 it would be very interesting to perform a comparison of HF and LF synchrony in predicted the EZ. Moreover, as mentioned below too, part of the HF synchrony can arise from trains of sharp spikes that have energy in all frequencies, which would not be synchrony of high frequency oscillations but an emphasis on the sharp part of ictal spikes. This is not a problem per se, but more an issue of interpretation of the results.22. Is it possible that part of the synchrony arises from filtered spikes? Can the authors provide a (normalized across frequencies) time-frequency representation of a representative seizure in order to assess the relative contribution of oscillations and spikes?

We appreciate your suggestion on using LF connectivity features to predict EZ and comparing the results with HF. While that can be a valuable investigation, we chose HF features based on the literature. Mainly, (Schevon et al., 2012) showed that core and penumbral might share similar low-frequency activity patterns. However, the high-frequency oscillations are the unique characteristics of the core. Importantly, our findings in Figures 4 and 6 in the main article presented that EZ-nR connectivity in mid-seizure has changed significantly for HF compared to the pre-ictal period. At the same time, this change was not significant in LF.

To investigate the contribution of ictal spikes and oscillations in SEEG signals, we computed the time-frequency representation (normalized across frequencies) of all signals in several seizures and patients. The following discussion and related figure (Figure 2—figure supplement 3) were added to the Results section of the manuscript.

“To investigate whether apparent high-frequency networks were produced as a result of ictal spikes rather than true oscillations, we examined the time-frequency representation of seizures. Figure 2—figure supplement 3 displays time-frequency plots for two sample seizures. The upper plot shows the spectrum averaged among nodes predicted as EZ, and the lower spectrum shows the case for randomly selected nodes outside the resection region, with the number of nodes in both groups equal. We observed pre-ictal spikes in the time-frequency plot for nodes inside the EZ, a previously reported feature of the epileptogenic zone (Grinenko et al., 2018). However, the figure shows that non-spiking high-frequency oscillations were the dominant activity after seizure onset. This observation supports our hypothesis that HFOs, rather than ictal spiking, are the main contributor to observed synchronization during seizures.”

23. P25, eigenvalue centrality. While the rank of the matrix is likely to be T, wouldn't one expect some kind of temporal correlation across points (arising from both the high overlap and some actual temporal persistence of the networks), which would result in an elbow in the eigenvalues after some dimension? If it is not the case, and each time point is independent from the other ones (linear eigenvalue spectrum), what is the added power of multilayer method? Wouldn't a more simple method as summing the degrees across time be equivalent? Please clarify.24. It would be interesting to visualize the eigenvalues of a representative example in order to measure this. Also, and importantly, it would be useful to compare the multilayer method to a simple mean degree across time (Wilke et al. 2010, van Mierlo et al. 2013, Courtens et al. 2016, Li et al. 2016, Balatskaya et al. 2020).

Previously in this response letter, we scrutinize the virtues of a multilayer framework in response to comments (#11, #13, and #14). We mentioned why a single-layer method cannot extract the neural processes with timescales slower than the time intervals between layers. Our EZ identification method showed that by increasing the coupling between layers we can explore across temporal scales.

We performed the requested analysis for the magnitude of the eigenvalues based on different coupling parameters. We inserted the following discussion and figure in the Results section:

“Figure 2—figure supplement 4 presents the changes in super-graph eigenvalues by adjusting the coupling parameter. For c≤1, there is a falloff when the number of eigenvalues (*n_e_*) meets the number of layers (*T*), indicating a super-adjacency matric with effective rank *T*. This is an expected result since the coupling is relatively small. When *c* increases, the eigenvalues become larger, and their corresponding eigenvectors would comprise several neighboring layers. Although the rate of decline in magnitude of eigenvalues accelerates as *c* increases, falloff can be detected when *n_e_* ≤ *2T*. By increasing the coupling value, the super-adjacency matrix transforms from a block diagonal matrix with effective rank *T,* to a matrix with major non-diagonal blocks and an effective rank far greater than *T*. In the computation of mlEVC, we considered the top *T* eigenvectors for all coupling parameters to avoid erroneous assumptions about the rank of super-graphs.”

References

Arnulfo, G., Wang, S.H., Myrov, V., Toselli, B., Hirvonen, J., Fato, M.M., Nobili, L., Cardinale, F., Rubino, A., Zhigalov, A., Palva, S., Palva, J.M., 2020. Long-range phase synchronization of high-frequency oscillations in human cortex. Nat. Commun. 11, 1–15. doi:10.1038/s41467-020-18975-8

Aydore, S., Pantazis, D., Leahy, R.M., 2013. A note on the phase locking value and its properties. Neuroimage 74, 231–244.

Bassett, D.S., Wymbs, N.F., Porter, M.A., Mucha, P.J., Carlson, J.M., Grafton, S.T., 2011. Dynamic reconfiguration of human brain networks during learning. Proc. Natl. Acad. Sci. U. S. A. 108, 7641–7646. doi:10.1073/pnas.1018985108

Betzel, R.F., Bassett, D.S., 2017. Multi-scale brain networks. Neuroimage 160, 73–83. doi:10.1016/j.neuroimage.2016.11.006

Bonacich, P., 1972. Factoring and weighting approaches to status scores and clique identification. J. Math. Sociol. 2, 113–120. doi:10.1080/0022250X.1972.9989806

Braun, U., Schäfer, A., Walter, H., Erk, S., Romanczuk-Seiferth, N., Haddad, L., Schweiger, J.I., Grimm, O., Heinz, A., Tost, H., Meyer-Lindenberg, A., Bassett, D.S., 2015. Dynamic reconfiguration of frontal brain networks during executive cognition in humans. Proc. Natl. Acad. Sci. U. S. A. 112, 11678–83. doi:10.1073/pnas.1422487112

Bullmore, E. and O.S., 2009. Complex brain networks: graph theoretical analysis of structural and functional systems. Nat Rev Neurosci. 10, 312. doi:10.1038/nrn2618

Burns, S.P., Santaniello, S., Yaffe, R.B., Jouny, C.C., Crone, N.E., Bergey, G.K., Anderson, W.S., Sarma, S. V., 2014. Network dynamics of the brain and influence of the epileptic seizure onset zone. Proc. Natl. Acad. Sci. 111, E5321–E5330. doi:10.1073/pnas.1401752111

Conrad, E.C., Bernabei, J.M., Kini, L.G., Shah, P., Mikhail, F., Kheder, A., Shinohara, R.T., Davis, K.A., Bassett, D.S., Litt, B., 2020. The sensitivity of network statistics to incomplete electrode sampling on intracranial eeg. Netw. Neurosci. 4, 484–506. doi:10.1162/netn_a_00131

De Domenico, M., Granell, C., Porter, M.A., Arenas, A., 2016. The physics of spreading processes in multilayer networks. Nat. Phys. 12, 901–906. doi:10.1038/nphys3865

Engel, J., da Silva, F.L., 2012. High-frequency oscillations – Where we are and where we need to go. Prog. Neurobiol. 98, 316–318. doi:10.1016/j.pneurobio.2012.02.001

Grinenko, O., Li, J., Mosher, J.C., Wang, I.Z., Bulacio, J.C., Gonzalez-Martinez, J., Nair, D., Najm, I., Leahy, R.M., Chauvel, P., 2018. A fingerprint of the epileptogenic zone in human epilepsies. Brain 141, 117–131. doi:10.1093/brain/awx306

Halkidi, M., 2001. On Clustering Validation Techniques – Springer 107–145.

Jacobs, J., Staba, R., Asano, E., Otsubo, H., Wu, J.Y., Zijlmans, M., Mohamed, I., Kahane, P., Dubeau, F., Navarro, V., Gotman, J., 2012. High-frequency oscillations (HFOs) in clinical epilepsy. Prog. Neurobiol. 98, 302–315. doi:10.1016/j.pneurobio.2012.03.001

Kahane, P., Landré, E., Minotti, L., Francione, S., Ryvlin, P., 2006. The Bancaud and Talairach view on the epileptogenic zone: a working hypothesis. Epileptic Disord. 8, 16–26.

Mucha, P.J., Richardson, T., Macon, K., Porter, M.A., Onnela, J.P., 2010. Community structure in time-dependent, multiscale, and multiplex networks. Science (80-. ). 328, 876–878. doi:10.1126/science.1184819

Nissen, I.A., van Klink, N.E.C., Zijlmans, M., Stam, C.J., Hillebrand, A., 2016. Brain areas with epileptic high frequency oscillations are functionally isolated in MEG virtual electrode networks. Clin. Neurophysiol. 127, 2581–2591. doi:10.1016/j.clinph.2016.04.013

Prichard, D., Theiler, J., 1994. Generating surrogate data for time series with several simultaneously measured variables. Phys. Rev. Lett. 73, 951.

Ray, S., Maunsell, J.H.R., 2011. Different origins of γ rhythm and high-γ activity in macaque visual cortex. PLoS Biol. 9. doi:10.1371/journal.pbio.1000610

Rubinov, M., Sporns, O., 2010. Complex network measures of brain connectivity: Uses and interpretations. Neuroimage 52, 1059–1069. doi:10.1016/j.neuroimage.2009.10.003

Schevon, C.A., Weiss, S.A., McKhann, G., Goodman, R.R., Yuste, R., Emerson, R.G., Trevelyan, A.J., 2012. Evidence of an inhibitory restraint of seizure activity in humans. Nat. Commun. 3, 1011–1060. doi:10.1038/ncomms2056

Smith, E.H., Liou, J.Y., Davis, T.S., Merricks, E.M., Kellis, S.S., Weiss, S.A., Greger, B., House, P.A., McKhann, G.M., Goodman, R.R., Emerson, R.G., Bateman, L.M., Trevelyan, A.J., Schevon, C.A., 2016. The ictal wavefront is the spatiotemporal source of discharges during spontaneous human seizures. Nat. Commun. 7, 1–12. doi:10.1038/ncomms11098

Tadel, F., Baillet, S., Mosher, J.C., Pantazis, D., Leahy, R.M., 2011. Brainstorm: A user-friendly application for MEG/EEG analysis. Comput. Intell. Neurosci. 2011, 1–13. doi:10.1155/2011/879716

Talairach, J., Bancaud, J., 1966. Lesion, “irritative” zone and epileptogenic focus. Confin. Neurol. 27, 91–94. doi:10.1159/000103937

Van Diessen, E., Hanemaaijer, J.I., Otte, W.M., Zelmann, R., Jacobs, J., Jansen, F.E., Dubeau, F., Stam, C.J., Gotman, J., Zijlmans, M., 2013. Are high frequency oscillations associated with altered network topology in partial epilepsy? Neuroimage 82, 564–573. doi:10.1016/j.neuroimage.2013.06.031

Wyllie, E., Gidal, B.E., Goodkin, H.P., Loddenkemper, T., Sirven, J.I., 2015. Wyllie’s treatment of epilepsy: principles and practice, 6th ed. Lippincott Williams & Wilkins.

Xu, Y., Yan, D., 2006. The Bedrosian identity for the Hilbert transform of product functions. Proc. Am. Math. Soc. 134, 2719–2728.

Zalesky, A., Fornito, A., Bullmore, E., 2012. On the use of correlation as a measure of network connectivity. Neuroimage 60, 2096–2106. doi:10.1016/j.neuroimage.2012.02.001

Zweiphenning, W.J.E.M., van ‘t Klooster, M.A., van Diessen, E., van Klink, N.E.C., Huiskamp, G.J.M., Gebbink, T.A., Leijten, F.S.S., Gosselaar, P.H., Otte, W.M., Stam, C.J., Braun, K.P.J., Zijlmans, G.J.M., 2016. High frequency oscillations and high frequency functional network characteristics in the intraoperative electrocorticogram in epilepsy. NeuroImage Clin. 12, 928–939. doi:10.1016/j.nicl.2016.09.014

[Editors' note: further revisions were suggested prior to acceptance, as described below.]

The manuscript has been improved but there are some remaining issues that need to be addressed, as outlined below. Please address these comments comprehensively in your response.1. In their response and amendments to the main text, the authors claim that their method does not require cross-validation and testing over independent datasets since the approach does not involve training the model parameters across patients and does not access information regarding the resection zone. While it is true that the method is applied to each patient, separately, the methodological choices including, but not limited to, functional connectivity metrics, frequency band selection, percentile threshold selection, still effectively "train" the general algorithm and may be highly specific to the cohort studied here. This issue presents itself as a driver of inter-patient variability, as demonstrated by the conflicting findings between the mlEVC method and Fingerprint method. While this limitation detracts from the methodological advances put forth by this study, but some caution in making the claim that cross-validation/testing is not necessary is warranted. Specifically, the manuscript should include a rationale for why cross-validation was not applied in this context, coupled with an acknowledgement that such an approach in future will be important to verify the current results.

We thank reviewers for their due diligence in examining the external validity of our algorithm.

Regarding the methodological choices, we would like to emphasize that the rationale for selecting the "connectivity metric" (lagged Coherence) was mentioned in the paper. Coherence is one of the most utilized connectivity metrics in neuroscience. It is also used in the article that we were inspired by (Burns et al., 2014) when performing this research. However, Coherence is prone to spurious connectivity due to volume conduction. Although, in comparison to scalp EEG, volume conduction is less of an issue in intracranial recordings, we were still interested in avoiding it in our study. As a result, we selected lagged Coherence in our analysis. The elementwise normalization of connectivity matrices with respect to the pre-ictal period was also followed by (Burns et al., 2014).

For "frequency band selection", the main question of our study was to investigate the brain connectivity in HFOs, as these oscillations are widely analyzed in epilepsy and are considered a biomarker of the epileptogenic zone. Since the sampling rate of the data was 500 Hz for several patients, we decided to choose the 80-200 Hz range to be consistent among patients. We then divided this band in half to have narrower frequency bands. Although we noticed that in some seizures or patients, only one band (80-140 Hz or 140-200 Hz) was informative, we did not drop any frequency band to avoid adding complexity to our algorithm. A future study can design an algorithm for tailoring the proper frequency band for each patient (Arnulfo et al., 2020).

Regarding our prediction and Fingerprint algorithm, we emphasized that the two methods use different features and can be complementary. Both algorithms have a small number of false positives, and the main differences are in electrodes predicted as true positives, which in both cases they were a subset of the electrodes in the resected area. Since both studies are performed retrospectively, the actual EZ is unknown. One probability can be that the union of the predicted regions by two algorithms is part of the actual EZ and its resection was necessary for seizure freedom among those patients.

Finally, we carefully edited our previous response and added more explanations to address the reviewers' concerns. These descriptions are written under a new section in the discussion titled as "limitations and considerations", which reads as follows:

"We should point out that cross-validation was not performed in this study for several reasons. First, resection labels were not used as part of the prediction algorithm, and the same performance function and ranges of coupling and thresholding values were applied to all subjects. Second, as shown in this manuscript and similar works, parameters such as coupling or active frequency band are patient-specific. By using cross-validation, we would have to fix these parameters based on a subset of patients, which may not be ideal as the actual range of these parameters could differ from the training dataset. While this study was primarily conducted to propose the multilayer network methodology during seizures, a more rigorous investigation is necessary to evaluate the external validity of our algorithm. Future research could use data from different clinical centers to assess the generalizability of the proposed technique and identify optimal parameters."

2. The relationship between the coupling parameter and network time-scale should be further elaborated, given the importance of this parameter to the authors' claim that patients may have different rates of network change. Please present data examining how sensitive the findings of the EZ are to the choice of this parameter. Please also provide examples of how different coupling parameters would yield more insight into the different time-scales of network change

We will address both comments at the same time. In summary, the reviewers requested that we quantify the performance of EZ identification if we used a single-layer model for all patients. Additionally, they asked whether employing different coupling values will result in a better understanding of brain dynamics.

We ran the EZ identification algorithm using both a single-layer model (with coupling set to 0) and a multilayer model with different coupling values (c = 1, … 10). We then compared these results to our proposed method, which involves analyzing a range of coupling parameters and weighting them based on clustering performance. Figure 2-source data 3 presents three performance measures for each case. These include the number of electrodes predicted as true positive, the number of electrodes predicted as false positive, and the number of patients with zero true positives. Our results demonstrate that the proposed approach yields superior EZ identification performance when compared to both the single-layer and multilayer models with fixed coupling values across all patients.

In addition to improving EZ identification, using the appropriate coupling parameter for each patient can also provide valuable insights into seizure dynamics. Author response image 4 compares the brain state dynamics during three seizures in two conditions: using c = 1 and the best-matched coupling parameter (c = 10 for this patient), as discussed in the main manuscript. With c = 1, there is no clear distinction between brain states, making it challenging to observe transitions between them. In contrast, when clustering is performed using the proper coupling parameter, the brain states become separable, allowing us to distinguish between neural state transitions. Although seizure semiology information was not available in our dataset, exploring its correlation with brain state transitions can reveal further insights into brain dynamics in future studies. Author response image 5 displays the same plot for patient 3.

**Author response image 4. sa2fig4:** Brain state transition during three seizures of patient 15 for two coupling values.

**Author response image 5. sa2fig5:** Brain state transition during three seizures of patient 3 for two coupling values.

3. The findings comparing a single-layer model to the multi-layer model should be explicitly quantified to concretely justify the claim in the Discussion section that single-layer models yielded poorer EZ identification than multi-layer models. How much more accurate was the multi-layer model to the single-layer model?4. More attention should be paid in the text to the risk of contamination of high frequency synchrony by two processes: (1) Filtering of spikes; and (2) Harmonics of non-sinusoidal oscillations. These topics are addressed in the following study, which could be cited in support of the discussion.Pitfalls of high-pass filtering for detecting epileptic oscillations: a technical note on "false" ripples. Bénar CG, Chauvière L, Bartolomei F, Wendling F. Clin Neurophysiol. 2010 Mar;121(3):301-105. The time frequency figures should be accompanied by a presentation of the signal time course in order to fully appreciate the presence/absence of spikes. While there indeed seems to be spikes in the preictal period (vertical lines) , they present much less energy than the high frequency activity during the seizure. This large increase is quite blurred and indistinct and it is not clear what this actually represents. The original signal time course would help understand which seizure pattern this corresponds to.A consideration of the potential effect of harmonics would also be helpful.

Author response image 6 shows the EEG signal of a single node predicted as the EZ, along with two time-frequency spectra. Two sections of this data are enlarged to better illustrate pre-ictal spikes and HFOs after seizure onset (Author response image 7).

As expected, the pre-ictal spikes display broadband activity across frequencies. In contrast, the HFOs, especially between 5-10s after onset, do not exhibit any spikes in the time domain and show a transparent "blob" in the frequency domain at 90-140 Hz. This frequency range falls within the first range we analyzed, but we did not observe significant activity in the 45-70 Hz range. These oscillations cannot be harmonics of non-sinusoidal components.

While we did not visually inspect the spectra in all seizures, our observations are consistent with the "ripple" described in the referenced article (Bénar et al., 2010).

Regarding the power comparison between HFOs and spikes, please note that we normalized these spectra for each frequency bin, as per your request. We performed this normalization to the pre-ictal period to better emphasize the changes that occur after the onset. Since there are stronger low-frequency components (such as spikes) before the onset, and more high-frequency oscillations after the seizure starts, these plots amplify the energy related to HFOs.

To address your concern, we added the following paragraph under the section of “limitations and considerations”:

“Filtering and subsequent analysis of HFOs should be performed with extra caution due to the possibility of producing false or spurious ripples. Research has shown that high-pass (> 80Hz) filtering of sharp transitions in EEG (such as artifacts and spikes) or harmonics of non-sinusoidal signals can result in such false ripples (Bénar et al., 2010). Several approaches have been suggested to investigate this phenomenon. These include visually inspecting the epileptic waveform alongside the filtered signal, using the time-frequency transform of the original signal since real and false ripples have different spectra, and employing automated spectral decomposition techniques such as Matching Pursuit (Bénar et al., 2010). In our analysis, we visually evaluated several seizures in both the time and time-frequency domains. However, it would be beneficial to use one of these mentioned algorithms in a systematic manner in future studies.”

**Author response image 6. sa2fig6:** EEG signal of one node inside the predicted EZ before and after seizure onset in the time domain (top), time-frequency using MATLAB (middle), and time-frequency using Brainstorm (bottom).

**Author response image 7. sa2fig7:** Pre-ictal spikes (top) and ictal HFOs (bottom) of one node inside the predicted EZ.